# Open-Vocabulary Customization from CLIP via Data-Free Knowledge Distillation

**Yongxian Wei[1], Zixuan Hu[2], Li Shen[3*], Zhenyi Wang[4], Chun Yuan[1*], Dacheng Tao[2]**
[1]Tsinghua University, China [2]Nanyang Technological University, Singapore
[3]Shenzhen Campus of Sun Yat-sen University, China [4]University of Maryland, College Park, USA

## Abstract

Vision-language models such as CLIP have demonstrated strong zero-shot performance, but their considerable size and inefficient inference limit customizable deployment for users. While knowledge distillation is a solution, it still requires the original data, which is not always available due to copyrights and privacy concerns. For many users seeking open-vocabulary customization, Data-Free Knowledge Distillation (DFKD) emerges as a promising direction. Upon rethinking DFKD, we find that existing methods fail on CLIP due to their heavy reliance on BatchNorm layers, which are unexpectedly *unusable* in CLIP. Based on our findings, we adopt image-text matching to achieve DFKD for CLIP, enabling customization based on arbitrary class texts. This involves (i) inversing a surrogate dataset from CLIP based on text prompts; and (ii) distilling a student model from CLIP using the surrogate dataset. Specifically, we introduce style dictionary diversification to enhance the *diversity* of synthetic images. To prevent uncontrollable semantics introduced by diversification, we propose a class consistency maintaining strategy to ensure the *consistency* of synthetic images. Based on synthetic images with various styles, we further propose meta knowledge distillation to train the student model with good *generalization* ability. Moreover, we introduce a simple yet effective method to enable customization based on few example images. Comprehensive experiments showcase the superiority of our approach across twelve customized tasks, achieving a 9.33% improvement compared to existing DFKD methods.

## 1 Introduction

Vision-Language Models (VLMs) such as CLIP (Radford et al., 2021) are revolutionizing the field with their outstanding performance. VLMs offer a remarkable zero-shot performance on a broad range of visual recognition tasks (Menon & Vondrick, 2023; Silva-Rodríguez et al., 2024) thanks to their open-vocabulary ability (Ilharco et al., 2022; Conti et al., 2023). However, the large model sizes, high computational resource requirements, and inefficient inference speed of these models restrict their deployment on mobile and IoT edge devices (Howard et al., 2017; Tan & Le, 2019). While knowledge distillation is a solution, it still requires the original data, which is not always available due to copyright and privacy concerns (Truong et al., 2021). Furthermore, different users have varying needs for downstream tasks, such as personalized tasks (Gal et al., 2023). This promotes us to ask: *Could we customize models based on users' needs (*e.g.*, arbitrary combinations of class texts or few example images) without the original data?* In other words, can we steal any compact models from an off-the-shelf CLIP to meet customized tasks?

Data-Free Knowledge Distillation (DFKD) (Yu et al., 2023; Hong et al., 2023) aims to distill a teacher model to a student model without accessing the original data. DFKD performs knowledge distillation by inversing a surrogate dataset from the teacher model. It typically optimizes the data $x$ associated with a target label $y$, which lies in the label space of the teacher model. However, applying SOTA methods to the powerful CLIP model results in significant failures (see Fig. 2). This is because existing DFKD relies on statistics stored in BatchNorm (BN) layers of the teacher model, which are used to align the distribution of synthetic images with the pre-training data distribution. However, the

---

*Corresponding author

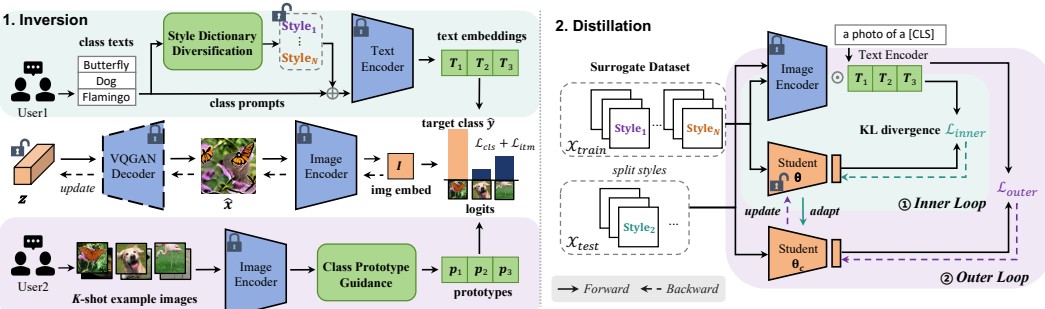

Figure 1: **An illustration of the open-vocabulary customization.** Users input their customized needs through class texts or example images and receive a student model that can directly perform inference. Our framework uses multi-modal encoders to build a loss function that evaluates how well a (text, image) or (image, image) pair matches, then backpropagates to the latent space. Diversity can be enhanced through style dictionary diversification. We optimize synthetic images to compile a surrogate dataset for training. Finally, we split images into training and testing sets according to styles, and employ meta-learning to train a customized student model.

BN statistics stored in CLIP are unexpectedly *unusable*. In this paper, we find that CLIP, trained on large-scale internet datasets, tends to encode *facial features* into its BN statistics (see Fig. 3).

To address the issue of unusable or absent BN layers (*e.g.*, architectures like ViT), we adopt an alternative inversion way via image-text matching (Xuan et al., 2023; Liu et al., 2024). By aligning the target class $t$ and synthetic image $x$ in the embedding space of CLIP, we achieve the reconstruction of original data (Liu et al., 2021; Crowson et al., 2022). However, there remain three drawbacks: (i) The diversity of synthetic images is insufficient, and they often lack photo-realism, failing to represent the real distribution; (ii) Diversifying text prompts to increase data diversity introduces uncontrollable semantics, potentially resulting in overly stylized and class-inconsistent content; (iii) Logits-based knowledge distillation tends to overfit by mimicking the teacher model's outputs, neglecting the invariant information and thus reducing generalization. Additionally, there is a lack of attention to how image prompts could also be utilized for inversion.

In this work, we address these issues in a unified framework (see Fig. 1), enabling customization based on both texts and images. We distill small models from CLIP in a data-free manner for customized tasks, obviating the need for the text encoder. For **text-based customization**, we introduce style dictionary diversification to enhance the diversity of synthetic images. Our style dictionary contains terms that describe the real world. We start with these terms as prompts and further utilize contrastive learning for instance-level discretization. This approach not only maintains photo-realism but also enhances diversity among samples. To prevent uncontrollable semantics introduced by diversification, we further propose class consistency maintaining to ensure accurate classification of synthetic images. After synthesizing images with various styles, we use meta knowledge distillation to transfer style-agnostic knowledge from CLIP to the student model. Specifically, we minimize the loss on current styles while ensuring that the optimization direction also yields improvements across other styles. By encouraging a gradient direction suitable for all styles, the student model captures shared representations across different styles, enabling effective generalization. For **image-based customization**, we construct prototypes based on example images to reduce intra-class variance. Instead of relying solely on few example images for knowledge distillation, we expand the distribution by leveraging the teacher's knowledge. We demonstrate that this approach effectively reduces the generalization error and enhances performance.

In summary, our main contributions are three-fold: (1) We enable open-vocabulary customization from CLIP based on only texts or few images. (2) We rethink model inversion methods in existing DFKD, uncovering the challenges posed by CLIP. (3) Our approach enhances the diversity and consistency of synthetic images, and improves the generalization of the student model.

## 2    RELATED WORK

**Vision-Language Models (VLMs).** VLMs (Jia et al., 2021; Li et al., 2022; 2023; Alayrac et al., 2022) have emerged as a promising paradigm for visual recognition with the release of CLIP (Radford et al., 2021). Zero-shot classification, *i.e.*, computing the similarity between a query image and

embedded texts for each class, has shown impressive results on various benchmarks. VLMs are trained on extensive datasets of image-text pairs. Unlike previous models limited to fixed classes, VLMs have the advantage of linking visual data with free-form language (Huang et al., 2024; Tang et al., 2024). Recent research has focused on optimizing input prompts, developing learnable prompts for text encoders (Zhou et al., 2022b;a), visual encoders (Bahng et al., 2022), or both jointly (Xing et al., 2023). Additionally, there's been explorations into using synthetic images generated from available class names to aid image classification (Udandarao et al., 2023; Bansal & Grover, 2023). In contrast, we do not query generative models or external images, but propose to inverse images directly from CLIP by optimizing images with pre-defined supervision.

**Data-Free Knowledge Distillation (DFKD).** DFKD (Raikwar & Mishra, 2022; Fang et al., 2022; Yu et al., 2023) aims to transfer knowledge from a pre-trained teacher model (typically a larger model) to a student model (generally more lightweight), without access to the raw data. This research is particularly valuable in practical scenarios where data availability is limited due to data privacy, safety, or ethical concerns (Hu et al., 2024; Wei et al., 2024b). DFKD has been greatly influenced by techniques like model inversion (Fredrikson et al., 2015), which seeks to extract embedded data knowledge from pre-trained models. In particular, DeepInversion (Yin et al., 2020) regularizes the distribution of synthetic images based on statistics from BN layers of the teacher model. Another method with a generator, CMI (Fang et al., 2021) introduces a contrastive learning objective to ensure that synthetic images are distinguishable from previously synthesized ones. They typically assume a classification model with a classifier that outputs logits. Target classes are represented as labels defined in the classifier's label space. In contrast, Xuan et al. (2023) proposed leveraging image-text matching for DFKD in VLMs, introducing three prompt diversification methods to extract out-of-distribution capacity from VLMs. In comparison, our target classes can be represented not only as texts but also as example images. Additionally, we propose consistency strategies to prevent noise semantics introduced by styles and enhance realism.

**Customization.** Adapting models to the specific needs of users has long been a goal in machine learning research. Customized models are typically seen in recommendation systems (Amat et al., 2018; Wang et al., 2023) and federated learning (Fallah et al., 2020; Atapour et al., 2024). Recently, the customization trend has extended to vision tasks, where fine-tuning generative models is common for reconstructing specific scenes (Roich et al., 2022; Gal et al., 2023; Kumari et al., 2023). PALAVRA (Cohen et al., 2022) utilizes a pre-trained CLIP model for the retrieval and segmentation of personalized objects. It identifies pseudo-words in the text embedding space of CLIP that refer to a specific object. These pseudo-words are then used to describe images for retrieval or to segment specific objects in a scene, distinguishing the personalized object from other candidates. In contrast, our focus lies in customizing smaller models that can directly perform inference to recognize objects.

## 3 PRELIMINARY

**Rethinking Model Inversion in DFKD.** Existing DFKD methods (Yin et al., 2020; Fang et al., 2021; 2022) typically employ a classification model that outputs logits, with target classes represented as labels defined in the classifier's label space. These methods focus on inversing images from pre-trained models using a classification loss and a prior regularization loss, treating the image $\hat{x}$ as the optimization object. The prior regularization loss aims to align the feature distribution of synthesized images with prior distribution information, *i.e.*, the mean $\mu$ and variance $\sigma^2$ stored in BN layers, encouraging the synthetic images $\hat{x}$ to mimic the distribution of original images:

$$\min_{\hat{x}} \mathcal{L}_{\text{BN}} = \sum_l \|\mu_l(\hat{x}) - \mu_l^{\text{BN}}\| + \|\sigma_l^2(\hat{x}) - \sigma_l^{\text{BN}}\|, \tag{1}$$

where $\mu_l(\hat{x})$ and $\sigma_l^2(\hat{x})$ denote the mean and variance of features at the $l^{\text{th}}$ convolutional layer. $\mu_l^{\text{BN}}$ and $\sigma_l^{\text{BN}}$ refer to the mean and variance of prior statistics stored in the $l^{\text{th}}$ BN layer.

To apply DFKD methods on VLMs, we utilize the ResNet-50 backbone of CLIP as the teacher model. However, CLIP only has a visual encoder $E_{\text{img}}$ to extract features. Consequently, we construct a linear classifier $W$ upon the backbone. We then fine-tune this classifier using the testing set to form a classification model $f(\cdot) = E_{\text{img}}(\cdot)^{\text{T}} W$, which replaces the teacher in DFKD methods.

As shown in Fig. 2, all DFKD methods fail when applied to CLIP. Detailed results for each dataset are provided in Table 8. This failure stems from their heavy reliance on BN layers, which store

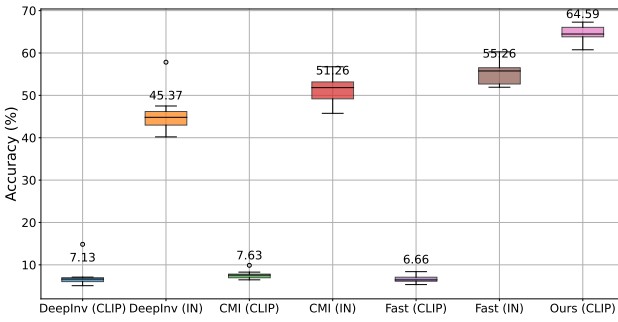

Figure 2: **Comparisons with DFKD methods on 11 customized tasks.** Average accuracies are reported. The notation (CLIP) refers to a teacher model of RN50 pre-trained by CLIP, and (IN) denotes a teacher model of RN50 pre-trained on ImageNet. Existing DFKD methods significantly fail with CLIP (single-digit accuracies across 100 classes), while only showing effectiveness with (IN). Our approach achieves the best results using CLIP.

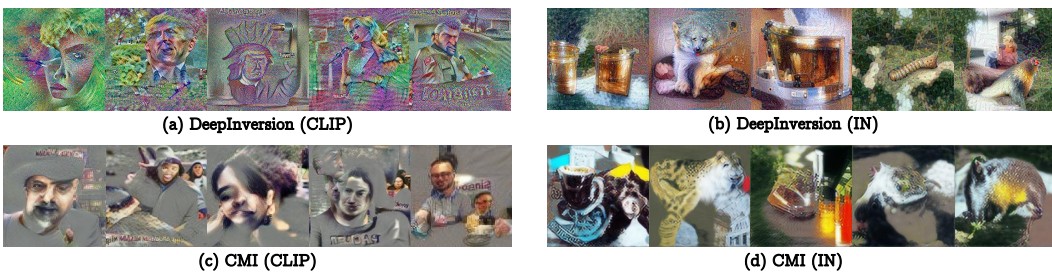

Figure 3: **Visualizations of synthetic images using DFKD methods.** We observe that CLIP, trained on large-scale internet datasets, tends to encode facial features into statistics of its BatchNorm layers. When using models pre-trained on ImageNet, these methods can synthesize informative images for training.

statistical knowledge from prior training. They are constrained to utilize the teacher pre-trained on training sets, which essentially stores the previously seen data. For instance, ablation experiments of DeepInversion (Yin et al., 2020) show a **40%-69%** drop in performance upon removal of the BN loss. While CLIP is pre-trained on large-scale, web-crawled datasets that often contain complex scenes with people, even though the text descriptions may not mention people (see Fig. 9 in App. B), leading to domain shifts from test images. We observe that CLIP's BN layers tend to favor faces (see Fig. 3), resulting in corrupted images that do not accurately reflect the target class. Other VLMs also exhibit this phenomenon, as detailed in App. B. Consequently, DFKD methods struggle to utilize CLIP to synthesize high-quality images, causing a significant performance drop.

To further validate this observation, we conduct experiments using a pre-trained ResNet-50 from PyTorch, combined with a linear classifier as the teacher. In Fig. 2, results denoted with (IN) clearly show improved performance, despite only fine-tuning the classifier. In Fig. 3, synthetic images (IN) can reflect class semantics and are recognizable by human observers. This suggests that **existing DFKD methods are only effective when the teacher's BN stored distribution closely matches the testing distribution**. Our approach enables DFKD for CLIP without relying on BN statistics, and is applicable even to transformers that lack BN layers, showcasing the superiority. Based solely on class texts, we outperform DFKD methods that utilize real data.

**Image-Text Matching.** To address the issue of unusable or absent BN layers (*e.g.*, architectures like ViT), we adopt an alternative inversion way via image-text matching. CLIP consists of a pair of text encoder $E_{\text{txt}}$ and image encoder $E_{\text{img}}$, which map a text $t$ and an image $x$ into a common latent space. Given class texts $\{t_1, \cdots, t_n\}$, we can synthesize images by aligning them in the latent space (Crowson et al., 2022; Liu et al., 2024). Specifically, we fix the CLIP parameters and adopt a pre-trained VQGAN with a decoder $\mathcal{G}$ for more efficient parameterization, conducting optimization directly in the latent variable space $z$. We start with random noise and progressively refine $z$ by minimizing the loss function, which measures image-text consistency:

$$\min_{z} \mathcal{L}_{itm} = \arcsin^2 \left( \| E_{\text{img}}(\mathcal{G}(z)) - E_{\text{txt}}(t) \| \right), \qquad (2)$$

where $\arcsin$ maps distances to angular space, and the image $\hat{x} = \mathcal{G}(z)$. The decoder $\mathcal{G}$ reduces the dimensionality of the optimization target from pixel space to a low-dimensional latent space.

To enhance image diversity, we create numerous prompt templates using a style dictionary $\{d_1, \cdots, d_N\}$, such as "[$t$] in the style of [$d$]", and introduce learnable styles optimized by instance-level contrastive learning (Xuan et al., 2023). Initially, input texts are converted into a set of tokens

Figure 4: **Left:** Utilizing image-text matching directly may result in synthetic images that lack photo-realism and tend towards artistic styles. **Right:** Introducing style dictionary diversification can lead to uncontrollable semantics, potentially resulting in overly stylized and class-inconsistent content. This may even cause CLIP itself to inaccurately recognize target class labels, imparting uncertain knowledge to the student model.

$V_t, \cdots, V_d$, where each word is mapped to a vector through a fixed codebook. These tokens are then fed into the transformer module to obtain the text embedding $T$. The contrastive loss is back-propagated to optimize $V_d$ of the style prompt $d$ in the token embedding space:

$$\min_{V_d} \mathcal{L}_{cst} = -\log \frac{\exp\left(d_{\cos}(T, T^+)\right)}{\sum_{b=1}^{B} \exp\left(d_{\cos}(T, T_b^-)\right)}, \tag{3}$$

where $d_{\cos}$ measures the distance between text embeddings, and $B$ denotes the batch size. Positive samples $T^+$ are obtained through text augmentation by randomly shuffling letters of the frozen template "in the style of". Negative samples $T_b^-$ are the other style prompts within the batch.

## 4 METHODOLOGY

In this section, we propose a data-free framework for customizing student models from CLIP (see Fig. 1). The framework adopts a two-step paradigm, including model inversion and knowledge distillation. In Sec. 4.1, we discuss text-based customization. Our approach enhances the diversity and consistency of synthetic images, and improves the generalization of the student model. In Sec. 4.2, we discuss customization based on few example images from each class.

### 4.1 TEXT-BASED CUSTOMIZATION

**Style dictionary diversification.** Given the target class $t$, we can synthesize images from CLIP by optimizing Eq. (3) and Eq. (2). Empirical observations (see Fig. 4, left) suggest that synthetic images often lack photo-realism and tend towards artistic styles. This phenomenon arises from CLIP's training on vast internet image-text pairs, containing various domains, enabling it to grasp abstract concepts. Hence, the style dictionary we devise aims to closely emulate real-world scenes, incorporating terms like "pattern", "illustration", and "photorealism". This strategy balances the maintenance of realism with the achievement of a diverse range of style prompts.

We can employ the concept of $\delta$-cover to investigate how data diversity influences the generalization error bound (Sener & Savarese, 2018; Zhang et al., 2023). A dataset $\mathcal{S}$ is a $\delta$-cover of dataset $\mathcal{D}$ if a set of balls of radius $\delta$, centered at samples in $\mathcal{S}$, cover the dataset $\mathcal{D}$. We define the diversity of the surrogate dataset $\mathcal{S}$ as $\delta_{\text{div}}$, which is the inverse of the minimal radius $\delta_{\min}$. The rationale is that if the radius $\delta_{min}$ is high, the diversity of the dataset must be low. Under some mild assumptions:

**Theorem 4.1.** *Given the original dataset $\mathcal{D} = \{x_i, y_i\}_{i\in[m]}$ with $m$ i.i.d. samples and the surrogate dataset $\mathcal{S} = \{\hat{x}_j, \hat{y}_j\}_{j\in[s]}$. Assume the hypothesis function is $\lambda^\eta$-Lipschitz continuous, the loss function $\ell(x, y)$ is $\lambda^\ell$-Lipschitz continuous for all $y$, and is bounded by $L$, with $\ell(\hat{x}_j, \hat{y}_j; \theta) = 0$ for all $j \in [s]$. If the dataset $\mathcal{S}$ is a $\delta$-cover of $\mathcal{D}$, with probability at least $1 - \gamma$, the bound holds:*

$$\left| \frac{1}{m} \sum_{i\in[m]} \ell(x_i, y_i; \theta) - \frac{1}{s} \sum_{j\in[s]} \ell(\hat{x}_j, \hat{y}_j; \theta) \right| \leq \frac{\lambda^\ell + \lambda^\eta LC}{\delta_{div}} + \sqrt{\frac{\log|\Theta| + \log\frac{1}{\gamma}}{2m}},$$

*where $C$ is the number of classes, and $\theta \in \Theta$ is the optimized student model.*

Please refer to App. A.1 for proofs. This theorem shows that the generalization error is bounded by $\delta_{div}$. In other words, the more diverse samples inversed from CLIP, the greater improvement in generalization can be achieved. Naive prompts result in a large covering radius $\delta$ and thus the $\delta_{div}$ is low. On the other hand, manually designing complex prompts is laborious and empirical. By

employing style dictionary diversification, CLIP can be guided to synthesize a diverse range of new samples, thereby enhancing the generalization of the student model.

**Class consistency maintaining.** It's inevitable that synthetic data may contain class-inconsistent samples (He et al., 2023). This issue becomes more severe with style dictionary diversification, as it may introduce uncontrollable semantics into text prompts (see Fig. 4, right). Hence, we introduce a class consistency maintaining strategy to prevent overly stylized deviation. Unlike the instance-by-instance optimization in Eq. (2), in a scenario where we know all the classes for the customized task, we can perform global multi-class optimization. Specifically, for an $n$-way classification, we input class names $\{t_1, \cdots, t_n\}$ with prompt $s_i =$ "a photo of a $[t_i]$" into the text encoder $E_{\text{txt}}$ to obtain text embeddings. Then, these text embeddings $E_{\text{txt}}(s)$ can be used to construct a classifier. Each latent embedding $E_{\text{img}}(\mathcal{G}(z))$ can be classified based on cosine similarity according to the classifier:

$$\min_{z} \mathcal{L}_{cls} = CE(E_{\text{img}}(\mathcal{G}(z)) \cdot E_{\text{txt}}(s), \hat{y}), \tag{4}$$

where $\hat{y} \in \mathbb{N}_{[1,n]}$ is the target class label. The classification loss serves as an anchor to regularize the class semantics of the diversified data in CLIP's embedding space. This promotes consistency between synthetic images and their corresponding class text.

**Meta knowledge distillation.** When using the surrogate dataset for distilling knowledge from CLIP, synthetic images may not cover all semantic information of real images, resulting in a gap between the training and testing distributions. In other words, the covariate shift issue (Sugiyama & Storkey, 2006) in DFKD is more significant than in knowledge distillation. To address this, we propose meta knowledge distillation to train the student model across various styles. The objective (Li et al., 2018) is to minimize the loss on current styles while ensuring that the optimization direction taken also yields improvements across other styles.

The surrogate dataset $\mathcal{X}$ contains $N$ styles through the style dictionary. To emulate real train-test shifts, we train the student model $\theta$ on different styles. During each iteration of training, one style $\mathcal{X}_{te}$ is randomly chosen as the meta-testing (virtual-test) style, while the remaining styles $\mathcal{X}_{tr} = \mathcal{X} - \mathcal{X}_{te}$ form the meta-training set. The optimization objective for meta knowledge distillation is as follows:

$$\min_{\theta} \mathbb{E}_{\mathcal{X}_{te} \sim \mathcal{X}} \mathcal{L}_{outer} = \mathcal{L}_{\text{KD}}(\mathcal{X}_{tr}; \theta) + \mathcal{L}_{\text{KD}}(\mathcal{X}_{te}; \theta_c), \text{ s.t. } \theta_c = \min_{\theta} \mathcal{L}_{inner} = \mathcal{L}_{\text{KD}}(\mathcal{X}_{tr}; \theta), \tag{5}$$

where $\mathcal{L}_{\text{KD}}$ is the KL divergence between student logits and CLIP logits, minimizing their disagreement. CLIP logits are the cosine similarity between image embeddings $E_{\text{img}}(\mathcal{X})$ and text embeddings $E_{\text{txt}}(s)$. $\theta_c$ is obtained after the inner loop, and $\theta$ is the student parameters we truly update. The outer loop evaluates the performance of $\theta_c$ on another style $\mathcal{X}_{te}$. To understand how the student learns invariant representations, we can analyze the objective from the perspective of gradient alignment.

**Theorem 4.2.** *If $\mathcal{L}_{\text{KD}}$ has Lipschitz Hessian, and $\theta_c = \theta - \alpha \nabla \mathcal{L}_{inner}$ denotes a single gradient descent step on $\theta$ with the objective $\mathcal{L}_{inner}$, where $\alpha$ is a learning rate, then:*

$$\nabla \mathcal{L}_{outer} = \nabla \mathcal{L}_{\text{KD}}(\mathcal{X}_{tr}; \theta) + \nabla \mathcal{L}_{\text{KD}}(\mathcal{X}_{te}; \theta) - \alpha \nabla \underbrace{(\nabla \mathcal{L}_{\text{KD}}(\mathcal{X}_{tr}; \theta) \cdot \nabla \mathcal{L}_{\text{KD}}(\mathcal{X}_{te}; \theta))}_{Style\ Alignment} + \mathcal{O}(\alpha^2).$$

Please refer to App. A.2 for proofs. The analysis reveals that the gradient of $\mathcal{X}_{te}$ produces the inner product with the gradient of $\mathcal{X}_{tr}$. This implies that optimizing the student model $\theta$ not only minimizes the expected loss across all styles $\mathcal{X}$ (effectively the empirical risk minimization), but also maximizes the inner product between gradients of different styles $\nabla \mathcal{L}_{KD}(\mathcal{X}_{tr}; \theta) \cdot \nabla \mathcal{L}_{KD}(\mathcal{X}_{te}; \theta)$. Hence, it encourages the student model to converge towards a common gradient direction across styles, learning invariant representations that generalize to new testing distributions.

## 4.2 IMAGE-BASED CUSTOMIZATION

In the above sections, we optimize synthetic images to match target texts, benefiting from CLIP's powerful image-text alignment capability. An interesting idea is whether we can optimize synthetic images using target images $\{I_1, \cdots, I_n\}$ by aligning their embeddings in the visual space, where each class has $K$-shot example images $\{i_j\}_{j=1}^{K}$. We can replace $E_{\text{txt}}(t)$ in Eq. (2) with $E_{\text{img}}(i)$ to enable the image prompt. Building upon this idea, we propose class prototype guidance to further reduce the intra-class variance of synthetic images.

A single image may not accurately capture the characteristics of a class, leading to synthetic images exhibiting notable intra-class variance, which will hinder inter-class discrimination (Hou & Sato, 2022). Therefore, we construct a prototype representation for each class, around which data points cluster. Specifically, we compute the prototype $\boldsymbol{p}_{\hat{y}}$ as the mean of few example images in the embedding space $E_{\text{img}}(\boldsymbol{i}_j)$. Synthetic images can be optimized towards the target class $\hat{y}$ by classifying it with all prototypes $\boldsymbol{p}$:

$$\min_{\boldsymbol{z}} \mathcal{L}_{iim} + \mathcal{L}_{cls} = \arcsin^2\left(\|E_{\text{img}}(\mathcal{G}(\boldsymbol{z})) - \boldsymbol{p}_{\hat{y}}\|\right) + CE(E_{\text{img}}(\mathcal{G}(\boldsymbol{z})) \cdot \boldsymbol{p}, \hat{y}). \tag{6}$$

Compared to solely using few example images to distill knowledge from the CLIP, our approach can be viewed as leveraging the CLIP's knowledge to augment a new distribution. We demonstrate that the generalization of a model trained on this distribution can be quantified by its proximity to the testing distribution. We borrow the notion of divergence from Ben-David et al. (2010).

**Corollary 4.3.** *For any arbitrary distributions $\mathcal{P}$ and $\mathcal{P}'$, with the trained model $\boldsymbol{\theta} \in \Theta$. The notion of $\Theta \Delta \Theta$ divergence denotes that $d_{\Theta \Delta \Theta}(\mathcal{P}', \mathcal{P}) = \sup_{\boldsymbol{\theta}, \boldsymbol{\theta}' \in \Theta} |\mathbb{P}_{\boldsymbol{x} \sim \mathcal{P}'}[\boldsymbol{\theta}(\boldsymbol{x}) \neq \boldsymbol{\theta}'(\boldsymbol{x})] - \mathbb{P}_{\boldsymbol{x} \sim \mathcal{P}}[\boldsymbol{\theta}(\boldsymbol{x}) \neq \boldsymbol{\theta}'(\boldsymbol{x})]|$. Assume there exists $\zeta(|\Theta|, s, \gamma) \geq 0$, a non-negative function that diminishes monotonically with $s$. Then, with probability at least $1 - \gamma$ the following bounds hold:*

$$E_{\mathcal{P}'}(\boldsymbol{\theta}) \leq \hat{E}_{\mathcal{P}}(\boldsymbol{\theta}) + \frac{1}{2} d_{\Theta \Delta \Theta}(\mathcal{P}', \mathcal{P}) + \zeta(|\Theta|, s, \gamma) + \lambda(\mathcal{P}'),$$

*where $E_{\mathcal{P}'}(\boldsymbol{\theta})$ is the expected error over $\mathcal{P}'$, $\hat{E}_{\mathcal{P}}(\boldsymbol{\theta})$ is the empirical error over the $s$ training samples in the surrogate dataset $\mathcal{S} \sim \mathcal{P}$, and $\lambda(\mathcal{P}')$ is a constant.*

Please refer to App. A.3 for proofs. The second term $d_{\Theta \Delta \Theta}(\mathcal{P}', \mathcal{P})$ represents the divergence between two distributions, underscoring that generalization performance depends on this divergence. Consequently, incorporating a distribution close to the testing distribution alongside the original data can significantly enhance generalization. This suggests that utilizing distributions generated from CLIP, a foundational model trained on a vast dataset, would be particularly beneficial due to its exposure to diverse distributions (Zhou et al., 2023).

## 5 EXPERIMENTS

In this section, we begin by detailing our experimental setup. Following that, we empirically verify the effectiveness of our proposed approach via extensive experiments.

### 5.1 EXPERIMENTAL SETUP

**Datasets and pre-trained models.** In the DFKD setting, we have no access to the training data. We employ ViT-B/32 as the visual encoder and a 12-layer 8-head transformer as the text encoder of CLIP (Radford et al., 2021). VQGAN (Esser et al., 2021) is utilized for synthesizing surrogate images. The pre-trained weights are kept fixed throughout the training process. We perform model inversion from texts sourced from datasets including Caltech-101 (Fei-Fei et al., 2004) (101 categories), ImageNet-1K (Deng et al., 2009) (1000 categories), or Flower-102 (Nilsback & Zisserman, 2008) (102 fine-grained categories). Our method is an open-vocabulary, customized approach suitable for any category recognized by CLIP. Therefore, we randomly divide ImageNet-1K into 10 splits to simulate a real customization scenario as closely as possible, reporting average results to demonstrate the robustness of our method. Each task includes over 100 categories encompassing a wide range of natural categories. Further details regarding data statistics are provided in App. H. The student model is evaluated on these datasets, including the test set of ImageNet, and the complete datasets of Caltech-101 and Flower-102, with the classification accuracy (in %) reported.

**Implementation details.** Without specific statements, we default to using a ResNet-18 as the student model. We construct a style dictionary with a size of 16 (details in App. G). The batch size for prompt learning is set to 64, with a learning rate of 0.01. Surrogate images are synthesized with a resolution of $224 \times 224$, and optimized using the Adam optimizer with a learning rate of 0.1 for 400 iterations. For text-based customization, 64 images are generated per class. For image-based customization, each class has 4 example images, and 24 additional images are synthesized per class. The inner loop learning rate $\alpha$ and outer loop learning rate for meta knowledge distillation are both set to 0.001, utilizing the SGD optimizer. Compared DFKD methods are implemented using the official code repository, and recommended hyperparameter settings are utilized.

Table 1: **Test accuracy (%) for text-based customization**. SDD: Style Dictionary Diversification; CCM: Class Consistency Maintaining; $\mathcal{L}_{CE}$: supervised loss with hard labels; $\mathcal{L}_{KD}$: knowledge distillation loss with soft labels; Meta: meta knowledge distillation. All inversion methods are tested with meta $\mathcal{L}_{KD}$.

| | | Caltech-101 | ImageNet1 | ImageNet2 | ImageNet3 | ImageNet4 | ImageNet5 | ImageNet6 | ImageNet7 | ImageNet8 | ImageNet9 | ImageNet10 | Average |
|---|---|---|---|---|---|---|---|---|---|---|---|---|---|
| T. | CLIP | 88.66 | 82.84 | 85.06 | 83.18 | 85.76 | 87.22 | 86.84 | 84.14 | 84.64 | 87.02 | 86.79 | 85.65 |
| Inversion | Baseline | 59.64 | 59.26 | 61.12 | 61.48 | 57.14 | 61.08 | 63.04 | 59.48 | 60.56 | 62.51 | 61.39 | 60.61 |
| | + SDD | 61.07 | **63.02** | 64.98 | 65.06 | 60.14 | 66.34 | **66.42** | 65.56 | 63.30 | 65.24 | **65.30** | 64.22 |
| | + CCM | 60.73 | 58.44 | 59.78 | 60.86 | 56.62 | 61.52 | 61.48 | 61.28 | 60.82 | 62.07 | 60.24 | 60.35 |
| | SDD+CCM | **61.33** | 62.46 | **65.02** | **65.60** | **62.52** | **66.80** | 66.24 | **66.78** | **65.62** | **65.49** | 65.00 | **64.81** |
| Distillation | $\mathcal{L}_{CE}$ | 55.19 | 54.28 | 58.70 | 58.82 | 54.88 | 59.69 | 61.92 | 59.92 | 57.14 | 58.39 | 56.73 | 57.79 |
| | $\mathcal{L}_{KD}$ | 59.90 | 61.56 | 63.76 | 64.34 | 60.66 | 65.33 | 65.48 | 65.62 | 64.22 | 64.27 | 64.22 | 63.58 |
| | $\mathcal{L}_{CE}+\mathcal{L}_{KD}$ | 59.87 | 60.14 | 62.42 | 63.82 | 58.54 | 64.07 | 64.22 | 63.82 | 63.84 | 62.77 | 62.29 | 62.35 |
| | Meta $\mathcal{L}_{CE+\mathcal{L}_{KD}}$ | 59.70 | 61.20 | 64.62 | 64.24 | 61.14 | 66.44 | 66.12 | 65.84 | 65.08 | 64.83 | 63.59 | 63.89 |
| | Meta $\mathcal{L}_{KD}$ | **61.33** | **62.46** | **65.02** | **65.60** | **62.52** | **66.80** | **66.24** | **66.78** | **65.62** | **65.49** | **65.00** | **64.81** |

## 5.2 MAIN RESULTS

**Results of text-based customization.** In Table 1, given the text $t$ of target classes, CLIP conducts zero-shot classification with the vanilla prompt "a photo of a [$t$]". Baseline (Crowson et al., 2022) synthesizes images using the vanilla prompt and performs meta knowledge distillation upon the dataset. Adding style dictionary diversification (SDD) notably improves performance, highlighting the importance of data diversity for knowledge distillation. SDD is a preliminary step with low complexity, taking only 57 seconds to train on RTX 4090. While SDD also introduces risks of noisy samples, *e.g.*, prompts may contain other confusing objects. Fortunately, with class consistency maintaining (CCM) to prevent overly stylized deviation, SDD+CCM yields consistent improvement over the baseline. It is observed that CCM sometimes brings performance drops compared with the baseline, which may be attributed to the reduction in diversity. Therefore, SDD+CCM is an effective trade-off, bringing an average performance improvement of 4.2%.

After obtaining the surrogate dataset, we explore various knowledge distillation methods. Compared to common supervised cross-entropy training, we find significant advantages in distilling knowledge from CLIP. This is because soft labels from the CLIP contain richer negative label information and higher entropy. Building upon this, we investigate the proposed meta knowledge distillation, which is an optimization approach that can be combined with different loss functions. "Meta" in Table 1 presents the effectiveness of our algorithm, where both $\mathcal{L}_{CE}+\mathcal{L}_{KD}$ and $\mathcal{L}_{KD}$ achieve improvements (↑1.54%). These results illustrate that meta knowledge distillation helps to discover shared class semantics across different styles and learn a more generalizable student.

**Results of image-based customization.** We explore image-based customization, where users have only few example images and may not even know the class names. As shown in Table 2, real data augmented with standard techniques such as RandomCrop, RandomFlip, and ColorJitter achieve certain effectiveness. However, Baseline (simply using each image as an image prompt for inversion) results in suboptimal performance, indicating that synthetic data is much less data-efficient than real data. On the other hand, our proposed prototype guidance boosts accuracy by 2.95%-4.76%. This confirms that CLIP effectively expands the training distribution while narrowing the divergence with the testing distribution, due to CLIP being trained on diverse distributions. As indicated by Corollary 4.3, approaching the testing distribution enhances generalization.

Table 2: **Image-based customization (%).**

| | Caltech-101 | ImageNet |
|---|---|---|
| Real data | 83.13 | 72.88 |
| Baseline | 81.83(-1.30%) | 69.78(-3.10%) |
| Ours | **84.78**(+1.65%) | **74.54**(+1.66%) |

**Computational complexity.** The CLIP model employs a ViT-B/32 visual encoder and a 12-layer 8-head text encoder, whereas the student model uses ResNet-18. As shown in Table 3, the student model not only has a more lightweight visual backbone but also omits the text encoder entirely. This leads to a significant reduction in both parameters and computational requirements.

Table 3: **Reductions in parameters and computational requirements compared to CLIP.**

|         | Img Params | Txt Params | Total Params | Img GFLOPs | Txt GFLOPs | Total GFLOPs |
|---------|------------|------------|--------------|------------|------------|--------------|
| CLIP    | 87.85M     | 63.43M     | 151.28M      | 8.82       | 5.96       | 14.78        |
| Student | 11.68M     | 0M         | 11.68M       | 1.82       | 0          | 1.82         |

Table 5: **Accuracies (%) on other student/teacher architectures** for knowledge distillation.

| Student | Teacher | Caltech-101 | ImageNet1 | ImageNet2 | ImageNet3 | ImageNet4 | ImageNet5 | ImageNet6 | ImageNet7 | ImageNet8 | ImageNet9 | ImageNet10 | Average |
|---------|---------|-------------|-----------|-----------|-----------|-----------|-----------|-----------|-----------|-----------|-----------|------------|---------|
| ViT-T   | RN50    | 58.48       | 57.64     | 59.16     | 60.82     | 59.34     | 58.04     | 59.68     | 57.64     | 58.35     | 60.30     | 60.22      | 59.06   |
| ViT-T   | ViT-B   | 55.17       | 56.38     | 57.14     | 57.42     | 56.52     | 60.05     | 61.64     | 60.48     | 59.54     | 60.55     | 58.61      | 58.50   |
| RN18    | RN50    | 60.76       | **63.90** | 64.52     | **66.40** | **63.18** | 66.46     | **67.28** | 64.06     | 63.74     | 64.48     | **65.71**  | 64.59   |
| RN18    | ViT-B   | **61.33**   | 62.46     | **65.02** | 65.60     | 62.52     | **66.80** | 66.24     | **66.78** | **65.62** | 65.49     | 65.00      | **64.81** |

## 5.3 FURTHER ANALYSES

**Trade-off between consistency and diversity.** The trade-off between class consistency maintaining and style dictionary diversification is critical. The hyperparameters controlling diversity are primarily the batch size and learning rate of contrastive learning. The hyperparameter controlling consistency is mainly the coefficient of Eq. (4), which we set to 1 by default. We conduct additional ablation studies by keeping the batch size constant and varying the learning rate to control diversity, as well as changing the loss coefficient to control consistency. The results are shown in Table 4. It can be seen that with the same diversity (fixed learning rate), increasing consistency shows beneficial trends. When consistency is the same (fixed coefficient), intermediate diversity generally performs better.

Table 4: **Consistency vs. diversity (%).**

| Coef. of $\mathcal{L}_{cls}$ | LR=0.001 | LR=0.01 | LR=0.1 |
|------------------------------|----------|---------|--------|
| 0.5                          | 61.64    | 62.10   | **63.22** |
| 1                            | 62.06    | 62.46   | 62.34  |
| 2                            | 62.35    | 62.94   | 62.52  |

**Knowledge distillation strategies.** Given the synthetic images, various training strategies have been explored in previous studies. For instance, He et al. (2023) utilizes each class's text features as the classifier's initialization, assuming it can aid convergence. Due to the initial instability of parameters and large gradients, directly learning the entire network may lead to numerical instability. Therefore, we start by training the classifier for warmup and then move on to train the whole model. Results of different strategies are shown in Fig. 5. Surprisingly, initializing with text features leads to a decrease in performance, indicating a discrepancy between the feature space of CLIP and the student model's feature space, which random initialization handles better. The effect of warmup is significant, as it helps alleviate premature overfitting to mini-batches in the initial stages, contributing to the stability of deeper layers in the student model.

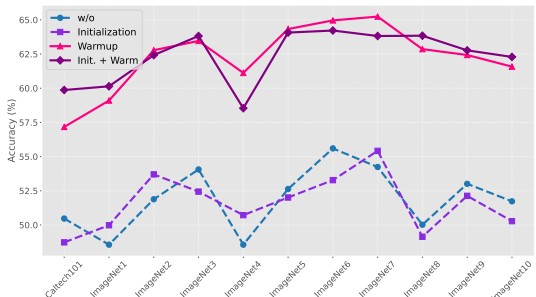

Figure 5: **Different training strategies.** All results use the $\mathcal{L}_{CE}+\mathcal{L}_{KD}$ for knowledge distillation.

**Other model architectures.** While our approach and theoretical framework are adapted for broad foundation models, we also explore its effectiveness on different model architectures, as shown in Table 5. We examine a different teacher architecture, CLIP-RN50. Additionally, we consider smaller student models like ViT-T (Touvron et al., 2021), which is a transformer architecture with a significantly reduced parameter count, enhancing its efficiency for image processing. Similar to other distillation techniques, we find that our approach is capable of transferring representations across various student/teacher architectures. This underscores the versatility of our approach: once synthesized, these datasets can be readily applied to train various model architectures.

**Visualizations.** We present visualized results of synthetic images generated by our approach. Specifically, Fig. 6 showcases the synthetic images produced based on class texts, while Fig. 7 displays those generated from example images. These visualized results illustrate the effectiveness of our style dictionary diversification in creating more diverse and plausible images, as well as the ability of image-based customization to generate content-consistent images.

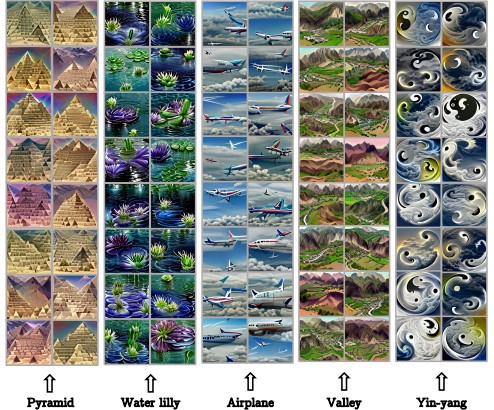

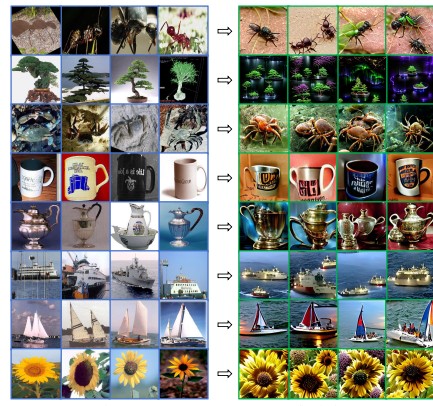

Figure 6: **Visualizations of the text-based customization.** Our approach enhances the diversity of synthesized images, preventing repetitive content while ensuring accurate representation of class semantics.

Figure 7: **Visualizations of the image-based customization.** Using only few example images, we synthesize corresponding images that serve as valuable supplements to the original images.

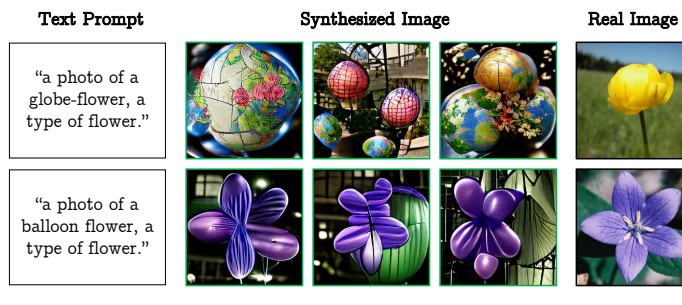

Figure 8: **Examples illustrating misalignment between input texts and synthesized images**, which can lead to suboptimal performance in knowledge distillation. CLIP may struggle with certain specialized terms and fine-grained classes. **Upper:** Request for a globe-flower, but CLIP ambiguously combines globe and flower. **Lower:** Balloon flower refers to a brand of flowers.

## 5.4 LIMITATION AND FUTURE WORK

In the experiments with Flower-102, a fine-grained flower dataset, we encounter limitations of CLIP inversion: incorrect synthesis of ambiguous text prompts. Short class names are inherently ambiguous (Song & Soleymani, 2019), and CLIP may misunderstand these texts or have not seen the class before, resulting in synthetic images that do not accurately represent the intended class (see Fig. 8). This issue arises not from our approach, but rather from a limitation inherent to VLMs, *i.e.*, the ambiguity of text prompts.

Table 6: **Strategies to alleviate text ambiguity (%).**

|  | Flower-102 |
| --- | --- |
| Text prompt | 15.83 |
| + Constraint | 18.07(+2.24%) |
| + Image prompt | 74.72(+58.89%) |

We outline several potential directions for future improvement: (1) Mitigating ambiguity using few example images, *i.e.*, combining both text and image prompts. (2) Employing more detailed text prompts for constraints, such as "a type of flower". Some studies (Menon & Vondrick, 2023; Pratt et al., 2023; Roth et al., 2023) have shown effectiveness in using LLMs to generate specific descriptions for each class. (3) Leveraging larger and more advanced VLMs (Li et al., 2021; 2022; 2023), with stronger text-image alignment capabilities, to address the limitation at the pre-training level. We conduct related experiments on Flower-102, and the results are shown in Table 6. Both image prompts and more detailed text prompts demonstrate their effectiveness. We also discuss the potential societal impacts of this work in App. F.

## 6 CONCLUSION

We rethink DFKD and find that existing methods fail on CLIP due to heavy reliance on BatchNorm layers, which are unexpectedly unusable in CLIP. In this paper, we enable open-vocabulary customization from CLIP based on only texts or few images. Our approach enhances the diversity and consistency of synthetic images, and improves the generalization of the student model. The effectiveness of each module is confirmed through theoretical analyses. Comprehensive experiments showcase the superiority of our approach across customized tasks.

## 7 ACKNOWLEDGMENTS

This work is supported by the National Key R&D Program of China (2022YFB4701400/4701402), SSTIC Grant (KJZD20230923115106012, KJZD20230923114916032, GJHZ20240218113604008), Beijing Key Lab of Networked Multimedia, the Shenzhen Basic Research Project (Natural Science Foundation) Basic Research Key Project (NO. JCYJ20241202124430041), the Major Project in Judicial Research from Supreme People's Court (NO. GFZDKT2024C08-3), CCF-DiDi GAIA Collaborative Research Funds (NO. CCF-DiDi GAIA 202419), and the National Research Foundation, Singapore, under its NRF Professorship Award No. NRF-P2024-001.

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

# A    THEORETICAL PROOFS

## A.1    PROOF FOR THEOREM 4.1

**Theorem A.1.** *Given the original dataset $\mathcal{D} = \{\boldsymbol{x}_i, y_i\}_{i\in[m]}$ with $m$ i.i.d. samples and the surrogate dataset $\mathcal{S} = \{\hat{\boldsymbol{x}}_j, \hat{y}_j\}_{j\in[s]}$. Assume the hypothesis function is $\lambda^\eta$-Lipschitz continuous, the loss function $\ell(\boldsymbol{x}, y)$ is $\lambda^\ell$-Lipschitz continuous for all $y$, and is bounded by $L$, with $\ell(\hat{\boldsymbol{x}}_j, \hat{y}_j; \boldsymbol{\theta}) = 0$ for all $j \in [s]$. If the dataset $\mathcal{S}$ is a $\delta$-cover of $\mathcal{D}$, with probability at least $1 - \gamma$, the bound holds:*

$$\left| \frac{1}{m} \sum_{i\in[m]} \ell(\boldsymbol{x}_i, y_i; \boldsymbol{\theta}) - \frac{1}{s} \sum_{j\in[s]} \ell(\hat{\boldsymbol{x}}_j, \hat{y}_j; \boldsymbol{\theta}) \right| \leq \frac{\lambda^\ell + \lambda^\eta LC}{\delta_{div}} + \sqrt{\frac{\log|\boldsymbol{\Theta}| + \log\frac{1}{\gamma}}{2m}},$$

*where $C$ is the number of classes, and $\boldsymbol{\theta} \in \boldsymbol{\Theta}$ is the optimized student model.*

*Proof.* Using PAC-Bayesian generalization bound (McAllester, 1999), for all $\boldsymbol{\theta} \in \boldsymbol{\Theta}$, with probability $1 - \gamma$ over independent draws $(\boldsymbol{x}_i, y_i) \sim \mathcal{P}_{\mathcal{X}\times\mathcal{Y}}$, we have:

$$\left| \mathbb{E}_{y_i\sim\eta(\boldsymbol{x}_i)}[\ell(\boldsymbol{x}_i, y_i; \boldsymbol{\theta})] - \frac{1}{m} \sum_{i\in[m]} \ell(\boldsymbol{x}_i, y_i; \boldsymbol{\theta}) \right| \leq \sqrt{\frac{\log|\boldsymbol{\Theta}| + \log\frac{1}{\gamma}}{2m}}.$$

Then, we reproduce and adapt the proof from Sener & Savarese (2018) in the context of core-set for completeness. We have a condition which states that there exists an $\hat{\boldsymbol{x}}_j$ in a $\delta$-cover around $\boldsymbol{x}_i$.

$$\begin{aligned}
\mathbb{E}_{y_i\sim\eta(\boldsymbol{x}_i)}[\ell(\boldsymbol{x}_i, y_i; \boldsymbol{\theta})] &= \sum_{k\in[C]} p_{y_i\sim\eta_k(\boldsymbol{x}_i)}(y_i = k)\ell(\boldsymbol{x}_i, k; \boldsymbol{\theta}) \\
&\overset{(d)}{\leq} \sum_{k\in[C]} p_{y_i\sim\eta_k(\hat{\boldsymbol{x}}_j)}(y_i = k)\ell(\boldsymbol{x}_i, k; \boldsymbol{\theta}) + \sum_{k\in[C]} |\eta_k(\boldsymbol{x}_i) - \eta_k(\hat{\boldsymbol{x}}_j)|\ell(\boldsymbol{x}_i, k; \boldsymbol{\theta}) \\
&\overset{(e)}{\leq} \sum_{k\in[C]} p_{y_i\sim\eta_k(\hat{\boldsymbol{x}}_j)}(y_i = k)\ell(\boldsymbol{x}_i, k; \boldsymbol{\theta}) + \delta\lambda^\eta LC,
\end{aligned}$$

where $y_i \sim \eta_k(\boldsymbol{x}_i)$ denotes that $\{y_i = k\} \sim \eta_k(\boldsymbol{x}_i) = p(y_i = k|\boldsymbol{x}_i)$. Step $(d)$ uses Claim 1 from Berlind & Urner (2015), and step $(e)$ uses the Lipschitz property of the hypothesis function and the loss bound. Considering the trained student model is assumed to have zero loss, *i.e.*, $\ell(\hat{\boldsymbol{x}}_j, \hat{y}_j; \boldsymbol{\theta}) = 0$.

$$\begin{aligned}
\sum_{k\in[C]} p_{y_i\sim\eta_k(\hat{\boldsymbol{x}}_j)}(y_i = k)\ell(\boldsymbol{x}_i, k; \boldsymbol{\theta}) &= \sum_{k\in[C]} p_{y_i\sim\eta_k(\hat{\boldsymbol{x}}_j)}(y_i = k)[\ell(\boldsymbol{x}_i, k; \boldsymbol{\theta}) - \ell(\hat{\boldsymbol{x}}_j, k; \boldsymbol{\theta})] \\
&\quad + \sum_{k\in[C]} p_{y_i\sim\eta_k(\hat{\boldsymbol{x}}_j)}(y_i = k)\ell(\hat{\boldsymbol{x}}_j, k; \boldsymbol{\theta}) \\
&\leq \delta\lambda^\ell.
\end{aligned}$$

Therefore, according to the definition of $\delta$-diversity, we have:

$$\mathbb{E}_{y_i\sim\eta(\boldsymbol{x}_i)}[\ell(\boldsymbol{x}_i, y_i; \boldsymbol{\theta})] \leq \frac{\lambda^\ell + \lambda^\eta LC}{\delta_{div}}.$$

Since we assume a zero training error for the surrogate dataset $\mathcal{S} = \{\hat{\boldsymbol{x}}_j, \hat{y}_j\}_{j\in[s]}$, applying the triangle inequality yields:

$$\begin{aligned}
|\frac{1}{m} \sum_{i\in[m]} \ell(\boldsymbol{x}_i, y_i; \boldsymbol{\theta}) - \frac{1}{s} \sum_{j\in[s]} \ell(\hat{\boldsymbol{x}}_j, \hat{y}_j; \boldsymbol{\theta})| &= \frac{1}{m} \sum_{i\in[m]} \ell(\boldsymbol{x}_i, y_i; \boldsymbol{\theta}) \\
&\leq \frac{\lambda^\ell + \lambda^\eta LC}{\delta_{div}} + \sqrt{\frac{\log|\boldsymbol{\Theta}| + \log\frac{1}{\gamma}}{2m}}.
\end{aligned}$$

$\square$

## A.2 PROOF FOR THEOREM 4.2

**Theorem A.2.** *If $\mathcal{L}_{\mathrm{KD}}$ has Lipschitz Hessian, and $\boldsymbol{\theta}_c = \boldsymbol{\theta} - \alpha\nabla\mathcal{L}_{inner}$ denotes a single gradient descent step on $\boldsymbol{\theta}$ with the objective $\mathcal{L}_{inner}$, where $\alpha$ is a learning rate, then:*

$$\nabla\mathcal{L}_{outer} = \nabla\mathcal{L}_{\mathrm{KD}}(\mathcal{X}_{tr};\boldsymbol{\theta}) + \nabla\mathcal{L}_{\mathrm{KD}}(\mathcal{X}_{te};\boldsymbol{\theta}) - \alpha\nabla\underbrace{(\nabla\mathcal{L}_{\mathrm{KD}}(\mathcal{X}_{tr};\boldsymbol{\theta}) \cdot \nabla\mathcal{L}_{\mathrm{KD}}(\mathcal{X}_{te};\boldsymbol{\theta}))}_{Style\ Alignment} + \mathcal{O}(\alpha^2).$$

*Proof.* We have:

$$\begin{aligned}
\nabla\mathcal{L}_{outer} =& \nabla\mathcal{L}_{\mathrm{KD}}(\mathcal{X}_{tr};\boldsymbol{\theta}) + \nabla\mathcal{L}_{\mathrm{KD}}(\mathcal{X}_{te};\boldsymbol{\theta}_c)\frac{\partial\boldsymbol{\theta}_c}{\partial\boldsymbol{\theta}} \\
=& \nabla\mathcal{L}_{\mathrm{KD}}(\mathcal{X}_{tr};\boldsymbol{\theta}) + \nabla\mathcal{L}_{\mathrm{KD}}(\mathcal{X}_{te};\boldsymbol{\theta}_c)\frac{\partial(\boldsymbol{\theta} - \alpha\nabla\mathcal{L}_{\mathrm{KD}}(\mathcal{X}_{tr};\boldsymbol{\theta}))}{\partial\boldsymbol{\theta}} \\
=& \nabla\mathcal{L}_{\mathrm{KD}}(\mathcal{X}_{tr};\boldsymbol{\theta}) + \nabla\mathcal{L}_{\mathrm{KD}}(\mathcal{X}_{te};\boldsymbol{\theta}_c)(\boldsymbol{I} - \alpha\nabla^2\mathcal{L}_{\mathrm{KD}}(\mathcal{X}_{tr};\boldsymbol{\theta})).
\end{aligned}$$

Applying the fundamental theorem of Taylor's expansion to the gradient $\nabla\mathcal{L}_{\mathrm{KD}}(\mathcal{X}_{te};\boldsymbol{\theta}_c)$, we have:

$$\begin{aligned}
& \nabla\mathcal{L}_{\mathrm{KD}}(\mathcal{X}_{te};\boldsymbol{\theta}_c) \\
=& \nabla\mathcal{L}_{\mathrm{KD}}(\mathcal{X}_{te};\boldsymbol{\theta}) + \nabla^2\mathcal{L}_{\mathrm{KD}}(\mathcal{X}_{te};\boldsymbol{\theta})(\boldsymbol{\theta}_c - \boldsymbol{\theta}) + \underbrace{\mathcal{O}(\|\boldsymbol{\theta}_c - \boldsymbol{\theta}\|^2)}_{=\mathcal{O}(\alpha^2)} \\
=& \nabla\mathcal{L}_{\mathrm{KD}}(\mathcal{X}_{te};\boldsymbol{\theta}) + \nabla^2\mathcal{L}_{\mathrm{KD}}(\mathcal{X}_{te};\boldsymbol{\theta})\underbrace{(\boldsymbol{\theta}_c - \boldsymbol{\theta})}_{-\alpha\nabla\mathcal{L}_{\mathrm{KD}}(\mathcal{X}_{tr};\boldsymbol{\theta})} + \mathcal{O}(\alpha^2) \\
=& \nabla\mathcal{L}_{\mathrm{KD}}(\mathcal{X}_{te};\boldsymbol{\theta}) - \alpha\nabla^2\mathcal{L}_{\mathrm{KD}}(\mathcal{X}_{te};\boldsymbol{\theta})\nabla\mathcal{L}_{\mathrm{KD}}(\mathcal{X}_{tr};\boldsymbol{\theta}) + \mathcal{O}(\alpha^2).
\end{aligned}$$

Note that, this offers a streamlined method to ascertain the Hessian Product by evaluating the gradient of $\mathcal{L}_{\mathrm{KD}}(\mathcal{X}_{te};\boldsymbol{\theta}_c)$ w.r.t. $\boldsymbol{\theta}_c$ (Wei et al., 2024c). As a result, this eliminates the necessity for the time-intensive and memory-consuming explicit calculation of the Hessian Product. Concurrently, any term that is a high-order term in $\alpha$ is classified into the $\mathcal{O}(\alpha^2)$ notation:

$$\begin{aligned}
\nabla\mathcal{L}_{outer} =& \nabla\mathcal{L}_{\mathrm{KD}}(\mathcal{X}_{tr};\boldsymbol{\theta}) + \nabla\mathcal{L}_{\mathrm{KD}}(\mathcal{X}_{te};\boldsymbol{\theta}) \\
& - \alpha\nabla^2\mathcal{L}_{\mathrm{KD}}(\mathcal{X}_{te};\boldsymbol{\theta})\nabla\mathcal{L}_{\mathrm{KD}}(\mathcal{X}_{tr};\boldsymbol{\theta}) - \alpha\nabla^2\mathcal{L}_{\mathrm{KD}}(\mathcal{X}_{tr};\boldsymbol{\theta})\nabla\mathcal{L}_{\mathrm{KD}}(\mathcal{X}_{te};\boldsymbol{\theta}) + \mathcal{O}(\alpha^2) \\
=& \nabla\mathcal{L}_{\mathrm{KD}}(\mathcal{X}_{tr};\boldsymbol{\theta}) + \nabla\mathcal{L}_{\mathrm{KD}}(\mathcal{X}_{te};\boldsymbol{\theta}) - \alpha\nabla\underbrace{(\nabla\mathcal{L}_{\mathrm{KD}}(\mathcal{X}_{tr};\boldsymbol{\theta}) \cdot \nabla\mathcal{L}_{\mathrm{KD}}(\mathcal{X}_{te};\boldsymbol{\theta}))}_{Style\ Alignment} + \mathcal{O}(\alpha^2).
\end{aligned}$$

$\square$

## A.3 PROOF FOR COROLLARY 4.3

**Corollary A.3.** *For any arbitrary distributions $\mathcal{P}$ and $\mathcal{P}'$, with the trained model $\boldsymbol{\theta} \in \boldsymbol{\Theta}$. The notion of $\boldsymbol{\Theta}\Delta\boldsymbol{\Theta}$ divergence denotes that $d_{\boldsymbol{\Theta}\Delta\boldsymbol{\Theta}}(\mathcal{P}',\mathcal{P}) = \sup_{\boldsymbol{\theta},\boldsymbol{\theta}'\in\boldsymbol{\Theta}}|\mathbb{P}_{\boldsymbol{x}\sim\mathcal{P}'}[\boldsymbol{\theta}(\boldsymbol{x}) \neq \boldsymbol{\theta}'(\boldsymbol{x})] - \mathbb{P}_{\boldsymbol{x}\sim\mathcal{P}}[\boldsymbol{\theta}(\boldsymbol{x}) \neq \boldsymbol{\theta}'(\boldsymbol{x})]|$. Assume there exists $\zeta(|\boldsymbol{\Theta}|,s,\gamma) \geq 0$, a non-negative function that diminishes monotonically with $s$. Then, with probability at least $1 - \gamma$ the following bounds hold:*

$$E_{\mathcal{P}'}(\boldsymbol{\theta}) \leq \hat{E}_{\mathcal{P}}(\boldsymbol{\theta}) + \frac{1}{2}d_{\boldsymbol{\Theta}\Delta\boldsymbol{\Theta}}(\mathcal{P}',\mathcal{P}) + \zeta(|\boldsymbol{\Theta}|,s,\gamma) + \lambda(\mathcal{P}'),$$

*where $E_{\mathcal{P}'}(\boldsymbol{\theta})$ is the expected error over $\mathcal{P}'$, $\hat{E}_{\mathcal{P}}(\boldsymbol{\theta})$ is the empirical error over the $s$ training samples in the surrogate dataset $\mathcal{S} \sim \mathcal{P}$, and $\lambda(\mathcal{P}')$ is a constant.*

*Proof.* From Theorem 2 in Ben-David et al. (2010), it holds that:

$$E_{\mathcal{P}'}(\boldsymbol{\theta}) \leq E_{\mathcal{P}}(\boldsymbol{\theta}) + \frac{1}{2}d_{\boldsymbol{\Theta}\Delta\boldsymbol{\Theta}}(\mathcal{P}',\mathcal{P}) + \lambda(\mathcal{P}'),$$

where $\frac{1}{2}d_{\boldsymbol{\Theta}\Delta\boldsymbol{\Theta}}(\mathcal{P}',\mathcal{P}) = \sup_{\boldsymbol{\theta},\boldsymbol{\theta}'\in\boldsymbol{\Theta}}|\mathbb{P}_{\boldsymbol{x}\sim\mathcal{P}'}[\boldsymbol{\theta}(\boldsymbol{x}) \neq \boldsymbol{\theta}'(\boldsymbol{x})] - \mathbb{P}_{\boldsymbol{x}\sim\mathcal{P}}[\boldsymbol{\theta}(\boldsymbol{x}) \neq \boldsymbol{\theta}'(\boldsymbol{x})]|$ and $\lambda(\mathcal{P}') = \min_{\boldsymbol{\theta}\in\boldsymbol{\Theta}} E_{\mathcal{P}'}(\boldsymbol{\theta}) + E_{\mathcal{P}}(\boldsymbol{\theta})$ is a constant. Then, with the PAC-Bayesian generalization bound:

$$\left|E_{\mathcal{P}}(\boldsymbol{\theta}) - \hat{E}_{\mathcal{P}}(\boldsymbol{\theta})\right| \leq \zeta(|\boldsymbol{\Theta}|,s,\gamma) = \sqrt{\frac{\log|\boldsymbol{\Theta}| + \log\frac{1}{\gamma}}{2s}},$$

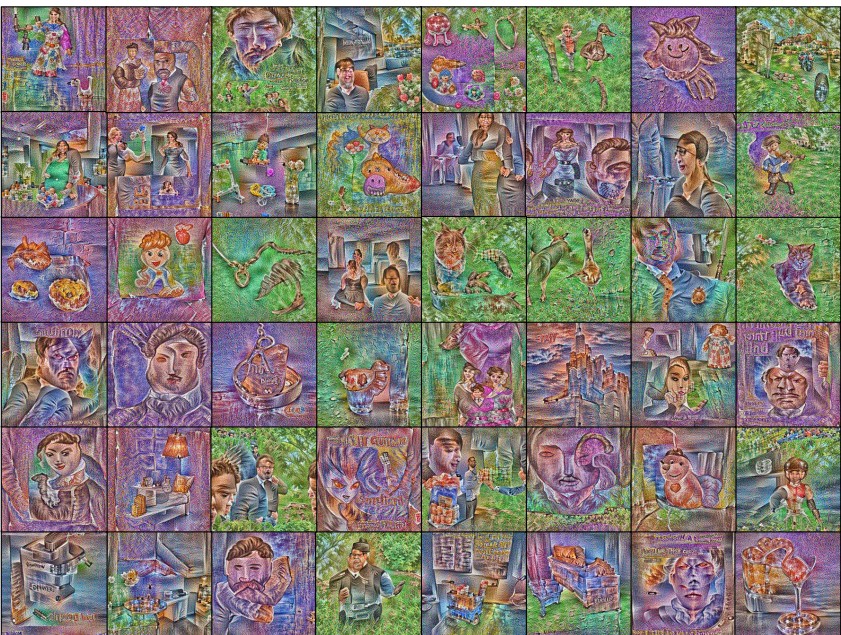

Figure 10: Visualizations of DeepInversion on Caltech-101 using PyramidCLIP (Gao et al., 2022).

with probability at least $1 - \gamma$, the following bounds hold:

$$E_{\mathcal{P}'}(\boldsymbol{\theta}) \leq \hat{E}_{\mathcal{P}}(\boldsymbol{\theta}) + \frac{1}{2}d_{\boldsymbol{\Theta}\Delta\boldsymbol{\Theta}}(\mathcal{P}', \mathcal{P}) + \zeta(|\boldsymbol{\Theta}|, s, \gamma) + \lambda(\mathcal{P}').$$

□

## B  RETHINKING DFKD ON OTHER VLMS

In Sec. 3, we examine the inversion results of existing DFKD methods on CLIP and find that almost all synthesized images are related to human faces, regardless of the target class. To explore whether other VLMs exhibit similar behavior, we present further inversion visualizations. However, the backbones of most VLMs are based on ViT, which lacks BN layers. This also highlights the superiority of our method, which remains effective even in the absence of BN layers. Consequently, we chose PyramidCLIP (Gao et al., 2022) (which provides a ResNet-50 backbone) for additional validation. PyramidCLIP addresses semantic mismatches and mutual compatibility issues with image-text pairs from the internet and outperforms CLIP by more than 10%, making it a stronger VLM.

Inversion results on PyramidCLIP are shown in Figs. 10 and 11. As depicted, half of the images successfully synthesize the target object, but the other half still depict human faces. This demonstrates a common phenomenon in VLMs: large-scale, web-crawled datasets invariably contain humans, even though the text descriptions may not mention people, leading to model bias (see Fig. 9). Consequently, existing DFKD methods are not applicable

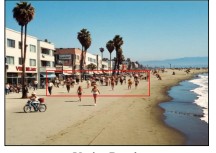
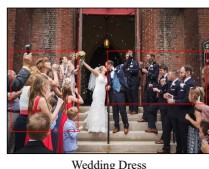

Venice Beach          Wedding Dress

Figure 9: **The problem in the web-crawled image-text pairs.**

to open-vocabulary foundation models and are limited to settings where the teacher model and testing set share the same distribution.

Our method is applicable to VLMs with a vision encoder and text encoder structure. We also explore the performance on BLIP (Li et al., 2022) and EVA (Fang et al., 2023). As shown in Table 7, EVA achieves the best performance, followed by CLIP and BLIP. The pre-trained weights we use for BLIP are the official "blip-itm-base-coco". We observe that BLIP's image-text matching capability is weaker than CLIP's. This is because BLIP's text encoder is image-grounded and trained on a binary classification task conditioned on images. Additionally, BLIP's pre-training dataset contains 14M

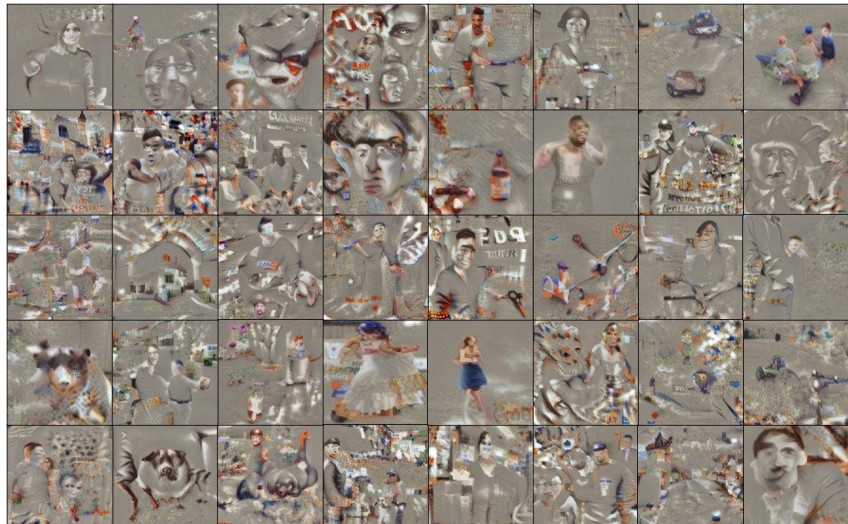

Figure 11: Visualizations of CMI on Caltech-101 using PyramidCLIP (Gao et al., 2022).

Table 7: **Test accuracy (%) for text-based customization across other VLMs.**

|  | Caltech-101 | ImageNet1 | ImageNet2 | ImageNet3 | **Average** |
|---|---|---|---|---|---|
| CLIP (Radford et al., 2021) | 61.33 | 62.46 | 65.02 | 65.60 | 63.60 |
| BLIP (Li et al., 2022) | 59.68 | 50.88 | 57.62 | 57.36 | 56.39 |
| EVA (Fang et al., 2023) | 67.62 | 65.24 | 66.96 | 66.42 | 66.56 |

images, whereas CLIP is pre-trained on 400M text-image pairs. EVA achieves the highest zero-shot top-1 accuracy on ImageNet-1K, and demonstrates strong image-text matching capability. These results indicate that our method is applicable to other VLM architectures and that its performance improves with the capability of the underlying VLM.

## C  WHY IMAGE-TEXT MATCHING EXCELS

While optimization in existing DFKD accurately captures the essence of a classification problem, such a formulation is sub-optimal for model inversion (Nguyen et al., 2023). In a classification setting, the primary expectation for $\hat{x}$ is to be sufficiently discriminative for class $i$. This objective can be achieved by both maximizing $E_{\text{img}}(\hat{x}) \cdot E_{\text{txt}}(t_i)$ or minimizing $\sum_{j=1, j \neq i}^{n} E_{\text{img}}(\hat{x}) \cdot E_{\text{txt}}(t_j)$. On the contrary, the goal of model inversion is to reconstruct training data. That is, in addition to $\hat{x}$ being sufficiently discriminative for class $i$, successful inversion also requires $\hat{x}$ to be close to the training data representations for class $i$ represented by $t_i$. Therefore, image-text matching focuses on maximizing $E_{\text{img}}(\hat{x}) \cdot E_{\text{txt}}(t_i)$ instead of maximizing the log-likelihood. Thanks to VLMs' powerful image-text alignment capability, we can reconstruct images close to the original data $x$.

## D  EXPERIMENT RESULTS

To compare our proposed text-based customization with existing DFKD methods (Yin et al., 2020; Fang et al., 2021; 2022), we summarize the results in Table 8. Conventional DFKD methods face challenges in leveraging CLIP's BN layers to generate high-quality images, leading to a notable performance drop. The specific results of different training strategies for knowledge distillation are presented in Table 9.

We also conduct hyperparameter sensitivity experiments for the inner and outer loop learning rates $\alpha$, used in meta knowledge distillation, as shown in Table 10. As the outer $\alpha$ increases, it leads to over-updating and significant performance drops. A lower step size allows the model to adapt more

Table 8: **Comparisons with DFKD methods on 11 customized tasks.** (CLIP) denotes a teacher model of ResNet-50 pre-trained by CLIP, and (IN) indicates a teacher model of ResNet-50 pre-trained on ImageNet.

| | Caltech-101 | ImageNet1 | ImageNet2 | ImageNet3 | ImageNet4 | ImageNet5 | ImageNet6 | ImageNet7 | ImageNet8 | ImageNet9 | ImageNet10 | Average |
|---|---|---|---|---|---|---|---|---|---|---|---|---|
| DeepInv (CLIP) | 14.84 | 6.26 | 6.24 | 5.08 | 6.76 | 5.70 | 7.06 | 6.64 | 7.10 | 6.87 | 5.84 | 7.13 |
| DeepInv (IN) | 57.83 | 41.50 | 44.98 | 44.46 | 40.92 | 47.40 | 40.20 | 44.54 | 44.82 | 47.49 | 44.92 | 45.37 |
| CMI (CLIP) | 9.88 | 7.88 | 7.74 | 6.92 | 7.58 | 7.72 | 8.28 | 6.46 | 7.54 | 6.92 | 6.98 | 7.63 |
| CMI (IN) | 56.75 | 49.30 | 51.82 | 53.36 | 45.76 | 51.92 | 49.06 | 52.98 | 47.22 | 55.67 | 50.04 | 51.26 |
| Fast (CLIP) | 5.34 | 7.38 | 6.16 | 6.44 | 7.02 | 5.98 | 7.14 | 7.00 | 6.28 | 8.39 | 6.10 | 6.66 |
| Fast (IN) | 60.26 | 55.76 | 55.98 | 57.02 | 51.92 | 52.86 | 56.02 | 52.50 | 52.04 | 57.91 | 55.63 | 55.26 |
| Ours (CLIP) | **60.76** | **63.90** | **64.52** | **66.40** | **63.18** | **66.46** | **67.28** | **64.06** | **63.74** | **64.48** | **65.71** | **64.59** |

Table 9: **Different training strategies.** All results use the $\mathcal{L}_{CE}+\mathcal{L}_{KD}$ for knowledge distillation.

| | Caltech-101 | ImageNet1 | ImageNet2 | ImageNet3 | ImageNet4 | ImageNet5 | ImageNet6 | ImageNet7 | ImageNet8 | ImageNet9 | ImageNet10 | Average |
|---|---|---|---|---|---|---|---|---|---|---|---|---|
| w/o | 50.47 | 48.56 | 51.88 | 54.06 | 48.56 | 52.63 | 55.60 | 54.24 | 50.02 | 53.01 | 51.73 | 51.89 |
| Initialization | 48.73 | 49.98 | 53.70 | 52.44 | 50.72 | 52.01 | 53.28 | 55.42 | 49.14 | 52.13 | 50.28 | 51.62 |
| Warmup | 57.17 | 59.10 | **62.78** | 63.46 | **61.12** | **64.33** | **64.96** | **65.24** | 62.86 | 62.43 | 61.58 | 62.28 |
| Init. + Warm | **59.87** | **60.14** | 62.42 | **63.82** | 58.54 | 64.07 | 64.22 | 63.82 | **63.84** | **62.77** | **62.29** | **62.35** |

finely to the common features of multiple tasks, rather than overfitting to any specific task. The best performance is achieved with an inner $\alpha$ of 0.01 and an outer $\alpha$ of 0.001.

TinyCLIP (Wu et al., 2023) and CLIP-KD (Yang et al., 2024) distill small CLIP models using relation, feature, gradient, and contrastive paradigms. Unlike our approach, they perform cross-modal distillation (students also have a text encoder) and rely on large-scale datasets such as LAION-400M, YFCC-15M, or CC3M+12M. LP-CLIP (Laroudie et al., 2023) trains a linear layer using pseudo-labels produced by CLIP, enhancing robustness through knowledge distillation on the training set. In contrast, we train the student network on synthetic data via meta-learning. We explore applying LP-CLIP's mixture of data augmentations and consistency loss to our student model distillation process. As shown in Table 11, consistency loss achieves comparable performance but slightly lags in generalization across different datasets.

# E  THE ROLE OF VQGAN

It replaces the statistics stored in the BN layers of pre-trained model to provide image priors (Struppek et al., 2022). This off-the-shelf generator aligns the patterns of synthetic images with natural images. The category of synthetic images is ensured through CLIP, as described in Eq. (2). Using an external generator offers more flexibility, allowing different pre-trained generators to be selected for the target task to minimize bias.

We conduct the following experiments: directly inverting images without VQGAN, using VQGAN pre-trained on ImageNet, and using VQGAN pre-trained on OpenImages. Additionally, we experimented with training VQGAN from scratch during the inversion process but encountered severe mode collapse. As shown in Table 12, VQGAN pre-trained on ImageNet outperforms OpenImages

Table 10: Effect of the learning rate in meta knowledge distillation on ImageNet1.

| | Outer $\alpha = 0.001$ | Outer $\alpha = 0.01$ | Outer $\alpha = 0.1$ |
|---|---|---|---|
| Inner $\alpha = 0.001$ | 62.46 | 56.88 | 5.44 |
| Inner $\alpha = 0.01$ | **63.58** | 56.66 | 6.22 |
| Inner $\alpha = 0.1$ | 62.04 | 55.82 | 10.68 |

Table 11: **Comparisons of different knowledge distillation techniques.**

|  | Caltech-101 | ImageNet1 | ImageNet2 | ImageNet3 | ImageNet4 | **Average** |
|---|---|---|---|---|---|---|
| Ours | 61.33 | **62.46** | **65.02** | **65.60** | 62.52 | **63.39** |
| LP-CLIP (Laroudie et al., 2023) | **61.40** | 60.50 | 62.34 | 64.68 | **63.94** | 62.57 |

Table 12: **VQGAN with different pre-trained weights.**

| Generator | ImageNet1 | ImageNet2 |
|---|---|---|
| Without VQGAN | 34.24 | 33.60 |
| ImageNet VQGAN | **62.46** | **65.02** |
| OpenImages VQGAN | 59.36 | 63.68 |

VQGAN when applied to ImageNet. The w/o VQGAN approach involves directly optimizing pixels for 400 iterations. While optimizing for thousands of iterations (as done in DeepInv) could improve results, this highlights the efficiency advantage of VQGAN.

We also considered using diffusion models as generators. Unlike VQGAN, which maps $z$ to the image $\hat{x}$ in a single forward pass, diffusion models require a multi-step denoising process to generate images from noisy inputs or latent variables. Even advanced samplers like DDIM (Song et al., 2021) or PNDM (Liu et al., 2022) typically require more than 10 steps. Performing model inversion involves optimizing input latent variables, which need to be differentiable. This means the full computation graph must be retained. However, most existing diffusion libraries (e.g., Hugging Face's Diffusers) use "$no\_grad()$" during generation. To address this, we implemented a differentiable version of DDIM for experimentation, but found that even with just 5 steps, the computation graph exceeded 24GB of GPU memory. In summary, we choose VQGAN for its efficiency and effectiveness.

## F    BROADER IMPACTS

Using large-scale image datasets poses significant challenges regarding privacy preservation, annotation labor, and AI ethics (Nakashima et al., 2022; Chen et al., 2024; Zhang et al., 2025). Consequently, large-scale datasets involving human-related images are being retracted due to ethical considerations. Other extensive datasets like JFT-300M and Instagram-3.5B (Mahajan et al., 2018) remain inaccessible to the public. These limitations significantly constrain research opportunities in this area, leading the research community to increasingly focus on the alternative use of pre-trained models (Zhang et al., 2024; Wei et al., 2024a).

In this context, Data-Free Knowledge-Distillation elegantly resolves these issues with open-sourced pre-trained models. By complying with the licensing terms set by the providers of these models, which may allow for academic, research, or commercial use, we can freely utilize them in downstream tasks. Furthermore, our approach offers practical solutions in resource-scarce scenarios, contributing to positive social impact by fulfilling users' customized needs.

## G    STYLE DICTIONARY

The initialized style dictionary consists of 16 style words, as depicted in Table 13. Based on empirical experience, these style words have shown good performance in real-world applications. We also experiment with a style dictionary containing styles sourced from the Internet and varied the number of styles. Generally, increasing the number of styles enhances diversity. However, as shown in Table 14, we observe a saturation trend where irrelevant styles may introduce negative bias. This occurs because some styles inadvertently add class-irrelevant semantics, such as unrelated elements or complex backgrounds. To address this, we carefully select style words that do not introduce additional semantics.

Table 13: Our style dictionary.

| photo | pattern | natural | picture |
|---|---|---|---|
| image | figure | profile | illustration |
| photorealism | realistic | digital | daguerreotype |
| expressionism | longexposure | isometric | view |

Table 14: **Accuracies (%) with differently initialized style dictionaries**.

|  | 16 styles | 50 styles | 86 styles | Our 16 styles |
|---|---|---|---|---|
| Caltech-101 | 57.99 | 60.78 | 60.49 | **61.33** |

## H   DATASET DETAILS

Caltech-101 = ['accordion', 'airplane', 'anchor', 'ant', 'barrel', 'bass', 'beaver',
'binocular', 'bonsai', 'brain','brontosaurus', 'buddha', 'butterfly', 'camera',
'cannon', 'car_side', 'ceiling_fan', 'cellphone', 'chair', 'chandelier', 'cougar_body',
'cougar_face', 'crab', 'crayfish', 'crocodile', 'crocodile_head', 'cup', 'dalmatian',
'dollar_bill', 'dolphin', 'dragonfly', 'electric_guitar', 'elephant', 'emu',
'euphonium', 'ewer', 'face', 'ferry', 'flamingo', 'flamingo_head','garfield', 'gerenuk',
'gramophone', 'grand_piano', 'hawksbill', 'headphone', 'hedgehog', 'helicopter',
'leopard', 'motorbike', 'ibis', 'inline_skate', 'joshua_tree', 'kangaroo', 'ketch',
'lamp', 'laptop', 'llama', 'lobster', 'lotus', 'mandolin', 'mayfly', 'menorah',
'metronome', 'minaret', 'nautilus', 'octopus', 'okapi', 'pagoda', 'panda', 'pigeon',
'pizza', 'platypus', 'pyramid', 'revolver', 'rhino', 'rooster', 'saxophone', 'schooner',
'scissors', 'scorpion', 'sea_horse', 'snoopy', 'soccer_ball', 'stapler', 'starfish',
'stegosaurus', 'stop_sign', 'strawberry', 'sunflower', 'tick', 'trilobite', 'umbrella',
'watch', 'water_lilly', 'wheelchair', 'wild_cat', 'windsor_chair', 'wrench', 'yin_yang']

ImageNet1 = ['leatherback sea turtle', 'bolo tie', 'perfume', 'sea slug', 'Soft-coated
Wheaten Terrier', 'eastern hog-nosed snake', 'Briard', 'radio', 'sliding door',
'cannon', 'horse chestnut seed', 'frilled-necked lizard', 'barbershop', 'wild boar',
'radiator grille', 'Dandie Dinmont Terrier', 'cowboy boot', 'thatched roof', 'beer
glass', 'catamaran', 'wallet', 'jellyfish', 'Clumber Spaniel', 'hen of the woods
mushroom', 'Irish Terrier', 'Sealyham Terrier', 'gondola', 'jeep', 'product packet _
packaging', 'pill bottle', 'paper towel', 'viaduct', 'geyser', 'coucal', 'pineapple',
'cardigan', 'otter', 'toy terrier', 'Schipperke', 'rotisserie', 'German Shorthaired
Pointer', 'monitor', 'baseball', 'hook', 'trombone', 'Cairn Terrier', 'Gordon Setter',
'ice cream', 'platypus', 'soda bottle', 'mongoose', 'guillotine', 'grey wolf', 'dung
beetle', 'dugong', 'Carolina anole', 'Leonberger', 'printer', 'Cocker Spaniel',
'Pekingese', 'electric ray', 'matchstick', 'waffle iron', 'window shade', 'cauldron',
'chain mail', 'hot dog', 'crayfish', 'Irish Water Spaniel', 'baby bib', 'French
Bulldog', 'marmoset', 'salt shaker', 'trench coat', 'black-footed ferret', 'chimpanzee',
'night snake', 'English foxhound', 'freight car', 'hay', 'common sorrel horse', 'common
redshank', 'sleeping bag', 'stretcher', 'threshing machine', 'rifle', 'conch', 'wool',
'Pickelhaube', 'block plane', 'minibus', 'fox squirrel', "yellow lady's slipper", 'Lhasa
Apso', 'lacewing', 'bow tie', 'vespa', 'Afghan Hound', 'prairie grouse', 'barbell']

ImageNet2 = ['go-kart', 'zebra', 'grey fox', 'throne', 'black-and-white colobus',
'apiary', 'swing', 'race car', 'CD player', 'moving van', 'sulphur butterfly', 'agaric',
'hunting bow', 'house finch', 'computer keyboard', 'Chesapeake Bay Retriever', 'eggnog',
'fur coat', 'tick', 'Shetland Sheepdog', 'sloth bear', 'snail', 'Mexican hairless
dog (xoloitzcuintli)', 'stingray', 'bee', 'tank', 'paintbrush', 'Giant Schnauzer',
'mud turtle', 'hummingbird', 'flagpole', 'tusker', 'great egret', "Geoffroy's spider
monkey", 'tea cup', 'pencil case', 'car wheel', 'automated teller machine', 'dough',
'half-track', 'southern black widow', 'water tower', 'tarantula', 'lakeshore', 'drink
pitcher', 'library', 'Saharan horned viper', 'brain coral', 'Basset Hound', 'leaf
beetle', 'one-piece bathing suit', 'sawmill', 'wok', 'sulphur-crested cockatoo',
'Yorkshire Terrier', 'rickshaw', 'sailboat', 'cherimoya (custard apple)', 'tailed frog',
'boathouse', 'mobile home', 'Flat-Coated Retriever', 'gibbon', 'saxophone', 'tree frog',
'sea cucumber', 'vine snake', 'semi-trailer truck', 'refrigerator', 'Japanese Chin',

'African rock python', 'goose', 'necklace', 'desk', 'disc brake', 'water jug', 'ground beetle', 'Border Terrier', 'cockroach', 'tights', 'great white shark', 'volcano', 'cliff dwelling', 'swim trunks _ shorts', 'patio', 'American coot', 'convertible', 'common gallinule', 'stethoscope', 'stopwatch', 'guacamole', 'shipwreck', 'European green lizard', 'space heater', 'manhole cover', 'tile roof', 'shoe store', 'steel drum', 'Papillon', 'totem pole']

ImageNet3 = ['fiddler crab', 'gyromitra', 'spatula', 'stinkhorn mushroom', 'rugby ball', 'armadillo', 'tandem bicycle', 'combination lock', 'rain barrel', 'picket fence', 'fire screen', 'slip-on shoe', 'brussels griffon', 'siamang', 'overskirt', 'ladle', 'espresso machine', 'great grey owl', 'plastic bag', 'Staffordshire Bull Terrier', 'eel', 'barrel', 'corn cob', 'Bloodhound', 'envelope', 'bullock cart', 'ladybug', 'airplane wing', 'Bouvier des Flandres dog', 'peafowl', 'through arch bridge', 'pedestal', 'Bullmastiff', 'trash can', 'American black bear', 'hare', 'Norwegian Elkhound', 'badger', 'missile', 'shower cap', 'hot tub', 'terrapin', 'Maltese', 'West Highland White Terrier', 'Golden Retriever', 'music speaker', 'whiskey jug', 'violin', 'fire salamander', 'four-poster bed', 'diaper', 'brown bear', 'black swan', 'snow leopard', 'bagel', 'Pomeranian', 'agama', 'pole', 'safe', 'cheetah', 'ocarina', 'jaguar', 'Shih Tzu', 'tiger beetle', 'grocery store', 'iPod', 'lab coat', 'Rottweiler', 'promontory', 'water buffalo', 'Bedlington Terrier', 'dome', 'notebook computer', 'syringe', 'screw', 'wall clock', 'traffic or street sign', 'howler monkey', 'barn', 'Greater Swiss Mountain Dog', 'daisy', 'poncho', 'chain-link fence', 'shoji screen _ room divider', 'balance beam', 'can opener', 'cradle', 'sunscreen', 'bustard', 'quilt', 'nematode', 'zucchini', 'wombat', 'borzoi', 'safety pin', 'greenhouse', 'spotted salamander', 'pipe organ', 'home theater', 'bolete']

ImageNet4 = ['solar thermal collector', 'hatchet', 'bassoon', 'orangutan', 'American Staffordshire Terrier', 'sombrero', 'oxygen mask', 'dock', 'mousetrap', 'hair wig', 'fly', 'gazelle', 'Vizsla', 'bald eagle', 'lynx', 'porcupine', 'knee pad', "potter's wheel", 'gar fish', 'boa constrictor', 'European polecat', 'electric guitar', 'lampshade', 'tow truck', 'sandbar', 'radiator', 'acorn', 'frying pan', 'muzzle', 'gong', 'honeycomb', 'mashed potatoes', 'airliner', 'drumstick', 'maraca', 'punching bag', 'spindle', 'meatloaf', 'mountain', 'Saluki', 'doormat', 'lighter', 'pool table', 'hornbill', 'reflex camera', 'airship', 'holster', 'Otterhound', 'artichoke', 'bulletproof vest', 'lifeboat', 'umbrella', 'red-breasted merganser', 'Siberian Husky', 'coffeemaker', 'graduation cap', 'earth star fungus', 'leafhopper', 'pillow', 'mixing bowl', 'microwave oven', 'cliff', 'Brittany dog', 'bobsleigh', 'car mirror', 'Christmas stocking', 'bottle cap', 'orange', 'Malinois', 'Indian cobra', 'baguette', 'backpack', 'jigsaw puzzle', 'parallel bars', 'partridge', 'container ship', 'Dungeness crab', 'lion', 'suit', 'Crock Pot', 'cardboard box _ carton', 'vacuum cleaner', 'coral fungus', 'digital clock', 'chocolate syrup', 'acoustic guitar', 'washing machine', 'tabby cat', 'sweatshirt', 'worm snake', 'English Setter', 'Labrador Retriever', 'gas pump', 'dhole', 'rock crab', 'fountain pen', 'toy store', 'Kerry Blue Terrier', 'dining table', 'dumbbell']

ImageNet5 = ['seat belt', 'Norfolk Terrier', 'drum', 'dust jacket', 'table lamp', 'Great Dane', 'Alaskan Malamute', 'Pembroke Welsh Corgi', 'china cabinet', 'vase', 'cornet', 'bubble', 'photocopier', 'hot pot', 'jackfruit', 'chainsaw', 'hammerhead shark', 'cauliflower', 'lipstick', 'sea snake', 'Entlebucher Sennenhund', 'sandal', 'ping-pong ball', 'crossword', 'teddy bear', 'Komodo dragon', 'cowboy hat', 'longhorn beetle', 'radio telescope', 'binoculars', 'goblet', 'golf cart', 'tram', 'corn', 'Groenendael dog', 'tripod', 'smooth newt', 'white-headed capuchin', 'tennis ball', 'comic book', 'bittern bird', 'sink', 'carbonara', 'gas mask or respirator', 'alligator lizard', 'ruddy turnstone', 'white stork', 'lotion', 'traffic light', 'oil filter', 'jacamar', 'amphibious vehicle', 'military hat (bearskin or shako)', 'military uniform', 'tray', 'marmot', 'aircraft carrier', 'Nile crocodile', 'upright piano', 'giant panda', 'echidna', 'king penguin', 'flamingo', 'chambered nautilus', 'St. Bernard', 'rock beauty fish', 'bassinet', 'high-speed train', 'CRT monitor', 'coffee mug', 'sea lion', 'harmonica', 'ocean liner', 'wardrobe', 'rooster', 'Dobermann', 'fountain', 'Toy Poodle', 'station wagon', 'suspension bridge', 'pufferfish', 'stupa', 'church', 'King Charles Spaniel', 'rocking chair', 'ruler measuring stick', 'hourglass', 'soap dispenser', 'Irish Setter', 'whistle', 'turnstile', 'American alligator', 'Alpine ibex', 'tiger', 'hockey puck', 'African wild dog', 'folding chair', 'vending machine', 'wooden spoon', 'analog clock']

```
ImageNet6 = ['scarf', 'clownfish', 'sidewinder rattlesnake', 'lorikeet', 'laptop
computer', 'cottontail rabbit', 'measuring cup', 'revolver', 'banana', 'baluster
_ handrail', 'coral reef', 'pinwheel', 'bathtub', 'jay', 'lawn mower', 'dog sled',
'chiton', 'bighorn sheep', 'box turtle', 'mosque', 'soup bowl', 'broom', 'snowplow',
'dishwasher', 'slug', 'balloon', 'ringlet butterfly', 'warthog', 'green iguana',
'Australian Silky Terrier', 'indri', 'sundial', 'Rhodesian Ridgeback', 'construction
crane', 'Redbone Coonhound', 'Australian Terrier', 'Dalmatian', 'abaya', 'recreational
vehicle', 'bridegroom', 'limpkin', 'barometer', 'strawberry', 'bakery', 'duck',
'dishcloth', 'payphone', 'tape player', 'poke bonnet', 'feather boa', 'cleaver', 'piggy
bank', 'acorn squash', 'electric locomotive', 'palace', 'Polaroid camera', 'dunlin',
'T-shirt', 'banded gecko', 'corkscrew', 'polar bear', 'academic gown', 'bra', 'English
Springer Spaniel', 'vulture', 'bell tower', 'dragonfly', 'Italian Greyhound', 'cello',
'scabbard', 'Ibizan Hound', 'pickup truck', 'tiger cat', 'cassette', 'red panda',
'messenger bag', 'hair clip', 'dowitcher', 'basketball', 'spiral or coil', 'pajamas',
'trifle', 'bell pepper', 'beer bottle', 'cucumber', 'crutch', 'small white butterfly',
'Beagle', 'ring binder', 'magnetic compass', 'mobile phone', 'sunglasses', 'quail',
'grasshopper', 'limousine', 'Band-Aid', 'green mamba', 'macaw', 'spider web', 'apron']

ImageNet7 = ['black stork', 'Samoyed', 'Great Pyrenees dog', 'Persian cat', 'Norwich
Terrier', 'Border Collie', 'ant', 'Arctic fox', 'chickadee', 'hamper', 'chameleon',
'parachute', 'smooth green snake', 'cougar', 'slot machine', 'snoek fish', 'Newfoundland
dog', 'bookcase', 'bulbul', 'Kuvasz', 'sneaker', 'shield', 'gown', 'lens cap', 'isopod',
'moped', 'hand-held computer', 'clothes iron', 'pelican', 'planetarium', 'motorboat',
'filing cabinet', 'pencil sharpener', 'prison', 'football helmet', 'pretzel', 'guenon',
'red wine', 'abacus', 'mountain bike', 'barber chair', 'cricket insect', 'ram
(adult male sheep)', 'railroad car', 'grand piano', 'baboon', 'pirate ship', 'mop',
'breastplate', 'tobacco shop', 'rotary dial telephone', 'German Shepherd Dog', 'baby
pacifier', 'weevil', 'oscilloscope', 'breakwater', 'maypole', 'kingsnake', 'Alaskan
tundra wolf', 'guinea pig', 'Asian elephant', 'chain', 'storage chest', 'Affenpinscher',
'French horn', 'plectrum', 'cassette player', 'cuirass', 'impala (antelope)', 'sea
anemone', 'brambling', 'sturgeon', 'bookstore', 'joystick', 'trimaran', 'stick insect',
'ring-necked snake', 'American robin', 'mink', 'hen', 'water bottle', 'grey whale',
'loupe magnifying glass', 'Treeing Walker Coonhound', 'monarch butterfly', 'megalith',
'velvet fabric', 'face powder', 'ballpoint pen', 'fig', 'shopping basket', 'website',
'yellow garden spider', 'European garden spider', 'Basenji', 'sock', 'trolleybus',
'weighing scale', 'toilet paper', 'little blue heron']

ImageNet8 = ['canoe', 'pig', 'electrical switch', 'tool kit', 'assault rifle', 'flute',
'sarong', 'school bus', 'stage', 'Australian Kelpie', 'pan flute', 'entertainment
center', 'wine bottle', 'spiny lobster', 'Old English Sheepdog', 'kit fox', 'sea
urchin', 'ring-tailed lemur', 'hammer', 'paddle', 'maze', 'red fox', 'altar', 'American
lobster', 'golf ball', 'mushroom', 'Sussex Spaniel', 'banjo', 'toaster', 'Boston
Terrier', 'junco', 'desktop computer', 'tiger shark', 'submarine', 'proboscis monkey',
'tractor', 'candle', 'plunger', 'coyote', 'rapeseed', 'desert grassland whiptail
lizard', 'wolf spider', 'Dutch oven', 'remote control', 'eraser', 'tricycle', 'wallaby',
'marimba', 'cheeseburger', 'Komondor', 'albatross', 'praying mantis', 'modem', 'paddle
wheel', 'lemon', 'teapot', 'Bernese Mountain Dog', 'cardoon', 'handkerchief', 'hyena',
'mortar and pestle', 'llama', 'medicine cabinet', 'killer whale', 'yurt', 'hair
dryer', 'front curtain', 'triumphal arch', 'Standard Schnauzer', 'Scottish Terrier',
'projector', 'soccer ball', 'padlock', 'park bench', 'carved pumpkin', 'purse', 'window
screen', 'dam', 'couch', 'scoreboard', 'bison', 'Gila monster', 'farm plow', 'pot pie',
'centipede', 'popsicle', 'Granny Smith apple', 'slide rule', 'knot', 'jeans', 'valley',
'goldfish', 'silver salmon', 'Siamese cat', 'oboe', 'oystercatcher', 'American dipper',
'neck brace', 'sewing machine', 'television']

ImageNet9 = ['african grey parrot', 'Whippet', 'beaver', 'hermit crab', 'shower
curtain', 'quill', 'toilet seat', 'Keeshond', 'Egyptian Mau', 'garter snake',
'black grouse', 'Bluetick Coonhound', 'Cardigan Welsh Corgi', 'strainer', 'Boxer',
'volleyball', 'odometer', 'police van', 'scorpion', 'red admiral butterfly', 'candy
store', 'milk can', 'carousel', 'mailbox', 'Appenzeller Sennenhund', 'plate', 'bath
towel', 'skunk', 'stone wall', 'baseball player', 'lionfish', 'kimono', 'croquet
ball', 'ambulance', 'pizza', 'vestment', 'pier', 'Chow Chow', 'schooner', 'mosquito
net', 'wheelbarrow', 'starfish', 'three-toed sloth', 'fishing casting reel', 'hard
disk drive', 'digital watch', 'keyboard space bar', 'spotlight', 'ostrich', 'Tibetan
Mastiff', 'vaulted or arched ceiling', 'gymnastic horizontal bar', 'military aircraft',
```

'red king crab', 'kite (bird of prey)', 'steam locomotive', 'hamster', 'microphone', 'pug', 'dingo', 'Petri dish', 'snorkel', 'arabian camel', 'titi monkey', 'cocktail shaker', 'ptarmigan', 'lighthouse', 'indigo bunting', 'snowmobile', 'flatworm', 'consomme', 'Chihuahua', 'hoop skirt', 'koala', 'Tibetan Terrier', 'American bullfrog', 'sports car', 'Black and Tan Coonhound', 'metal nail', 'Welsh Springer Spaniel', 'birdhouse', 'red wolf or maned wolf', 'Miniature Poodle', 'trilobite', 'horse-drawn vehicle', 'loggerhead sea turtle', 'mask', 'parking meter', 'torch', 'harp', 'African bush elephant', 'power drill', 'eastern diamondback rattlesnake', 'Standard Poodle', 'fire truck', 'clogs', 'drilling rig', 'plant pot', 'bikini', 'cicada']

ImageNet10 = ['meerkat', 'weasel', 'combine harvester', 'newt', 'water snake', 'obelisk', 'crash helmet', 'hippopotamus', 'pomegranate', 'butcher shop', 'tent', 'Angora rabbit', 'unicycle', 'Scottish Deerhound', 'Miniature Schnauzer', 'accordion', 'space shuttle', 'Airedale Terrier', 'common squirrel monkey', 'forklift', 'ford model t', 'racket', 'computer mouse', 'husky', 'cabbage', 'burrito', 'gorilla', 'taxicab', 'magpie', 'prayer rug', 'bell or wind chime', 'Windsor tie', 'Wire Fox Terrier', 'ski', 'miniskirt', 'plate rack', 'broccoli', 'scuba diver', 'mitten', 'Curly-coated Retriever', 'brass memorial plaque', 'patas monkey', 'garbage truck', 'tench', 'split-rail fence', 'shovel', 'electric fan', 'restaurant', 'shopping cart', 'Lakeland Terrier', 'espresso', 'stove', 'swimming cap', 'barn spider', 'Irish Wolfhound', 'typewriter keyboard', 'damselfly', 'triceratops', 'movie theater', 'leopard', 'buckle', 'fireboat', 'hair spray', 'chiffonier', 'menu', 'beach', 'butternut squash', 'axolotl', 'Miniature Pinscher', 'crate', 'letter opener', 'goldfinch', 'ox', 'collie', 'spaghetti squash', 'castle', 'gossamer-winged butterfly', 'thimble', 'harvestman', 'Weimaraner', 'macaque', 'beaker', 'spoonbill', 'monastery', 'rhinoceros beetle', 'minivan', 'rose hip', 'ruffed grouse', 'hartebeest', 'balaclava ski mask', 'crane bird', 'bucket', 'screwdriver', 'bee eater', 'toucan', 'infant bed', 'langur', 'cloak']

Flower-102 = ['pink primrose', 'hard-leaved pocket orchid', 'canterbury bells', 'sweet pea', 'english marigold', 'tiger lily', 'moon orchid', 'bird of paradise', 'monkshood', 'globe thistle', 'snapdragon', "colt's foot", 'king protea', 'spear thistle', 'yellow iris', 'globe-flower', 'purple coneflower', 'peruvian lily', 'balloon flower', 'giant white arum lily', 'fire lily', 'pincushion flower', 'fritillary', 'red ginger', 'grape hyacinth', 'corn poppy', 'prince of wales feathers', 'stemless gentian', 'artichoke', 'sweet william', 'carnation', 'garden phlox', 'love in the mist', 'mexican aster', 'alpine sea holly', 'ruby-lipped cattleya', 'cape flower', 'great masterwort', 'siam tulip', 'lenten rose', 'barbeton daisy', 'daffodil', 'sword lily', 'poinsettia', 'bolero deep blue', 'wallflower', 'marigold', 'buttercup', 'oxeye daisy', 'common dandelion', 'petunia', 'wild pansy', 'primula', 'sunflower', 'pelargonium', 'bishop of llandaff', 'gaura', 'geranium', 'orange dahlia', 'pink-yellow dahlia', 'cautleya spicata', 'japanese anemone', 'black-eyed susan', 'silverbush', 'californian poppy', 'osteospermum', 'spring crocus', 'bearded iris', 'windflower', 'tree poppy', 'gazania', 'azalea', 'water lily', 'rose', 'thorn apple', 'morning glory', 'passion flower', 'lotus', 'toad lily', 'anthurium', 'frangipani', 'clematis', 'hibiscus', 'columbine', 'desert-rose', 'tree mallow', 'magnolia', 'cyclamen', 'watercress', 'canna lily', 'hippeastrum', 'bee balm', 'ball moss', 'foxglove', 'bougainvillea', 'camellia', 'mallow', 'mexican petunia', 'bromelia', 'blanket flower', 'trumpet creeper', 'blackberry lily']

