# OpenReview forum: "Open-Vocabulary Customization from CLIP via Data-Free Knowledge Distillation"
_ICLR.cc/2025/Conference — ICLR 2025 Oral_

### Official Review · Reviewer_Radj · 2024-11-04

**Soundness:** 3
**Presentation:** 3
**Contribution:** 3
**Rating:** 8
**Confidence:** 4

**Summary:**

This paper addresses the customization of CLIP for specific user-defined tasks without using original data. The proposed approach involves generation of synthetic images using VQGAN in different styles to increase diversity, while following a data-free meta learning based knowledge distillation technique to adapt a lioghtweight student encoder from teacher CLIP. It aims to overcome the reliance on BatchNorm layers, which hinder customization for ViT variants of CLIP model. The authors have shown extensive experiments with significant improvement of performance of the proposed method compared to CLIP.

**Strengths:**

1. The method eables model customization without accessing the original data, preserving privacy of the users.
2. The proposed approach captures invariant representations through style diversification and meta knowledge distillation, which is interesting.

**Weaknesses:**

1. As CLIP is already a very good domain-aware model, what is the motivation behind generating style tranferred images? The diversification could be better and challenging with generation of very fine-grained realistic images.

2. Can pretrained diffusion models be used instead of VQGAN, as it can generate more diverse datasets very easily? What are the pros and cons of using a diffusion model?

3. Why meta learning based knowledge distillation over traditional supervised learning? Any theorectical reason?

    experiments of distillation techniques like TinyCLIP [1], CLIP-KD [2], LP-CLIP [3] are likely to be preferable.


    [1] TinyCLIP: CLIP Distillation via Affinity Mimicking and Weight Inheritance, ICCV 2023.

    [2] CLIP-KD: An Empirical Study of CLIP Model Distillation, CVPR 2024.

    [3] Improving CLIP Robustness with Knowledge Distillation and Self-Training

**Questions:**

See the weakness section. I would like to increase my rating, if the proper justification of my questions will be given.

---

> ### Author Response · Authors · 2024-11-19
> **Author Rebuttal (1/1)**
>
> > Q1: As CLIP is already a very good domain-aware model, what is the motivation behind generating style tranferred images?
>
> A1: It is precisely because CLIP is a very good domain-aware model that we can invert images from it with diverse styles. **The diversity of training data is crucial for effective knowledge distillation.** We analyze the generalization error of the inversed data through Theorem 4.1, providing a theoretical explanation for the benefits of diversification. We also prioritize generating realistic images. The style dictionary we devise aims to closely emulate real-world scenes.
>
> As noted in [4], ensembling multiple prompts, including even random words, can improve CLIP’s performance. This has been interpreted as a form of noise augmentation, which enhances the robustness of model for a variety of domains.
> ___
> > Q2: Can pretrained diffusion models be used instead of VQGAN, as it can generate more diverse datasets very easily? What are the pros and cons of using a diffusion model?
>
> A2: We also considered using diffusion models as generators. Unlike VQGAN, which maps $\boldsymbol{z}$ to the image $\hat{\boldsymbol{x}}$ in a single forward pass, diffusion models require a multi-step denoising process to generate images from noisy inputs or latent variables. Even advanced samplers like DDIM [5] or PNDM [6] typically require more than 10 steps. Performing model inversion involves optimizing input latent variables, which need to be **differentiable**. This means the full computation graph must be retained. However, most existing diffusion libraries (e.g., Hugging Face's `Diffusers`) use `no_grad()` during generation. To address this, we implemented a differentiable version of DDIM for experimentation, but found that even with just 5 steps, the computation graph exceeded 24GB of GPU memory. In summary, we choose VQGAN for its efficiency and effectiveness.
> ___
> > Q3: Why meta learning based knowledge distillation over traditional supervised learning? Any theorectical reason?
>
> A3: There are two reasons for using meta knowledge distillation:
> * "Logits-based knowledge distillation tends to overfit by mimicking the teacher model's outputs, neglecting the invariant information and thus reducing generalization." (lines 77-78)
> * "When using the surrogate dataset for distilling knowledge from CLIP, synthetic images may not cover all semantic information of real images, resulting in a gap between the training and testing distributions. In other words, the covariate shift issue in DFKD is more significant than in knowledge distillation." (lines 284-287)
>
> Generalization is crucial when working with synthetic datasets, and it benefits from the implicit gradient inner product of meta-learning [7]. Meta knowledge distillation minimizes the loss on current styles while ensuring that the optimization direction also yields improvements across other styles. By encouraging a gradient direction suitable for all styles, the student model learns shared representations across different styles. We have demonstrated its effectiveness experimentally (+1.54%) and analyzed the reasons for its effectiveness in Theorem 4.2.
>
> TinyCLIP [1] and CLIP-KD [2] distill small CLIP models using relation, feature, gradient, and contrastive paradigms. Unlike our approach, they perform cross-modal distillation (students also have a text encoder) and rely on large-scale datasets such as LAION-400M, YFCC-15M, or CC3M+12M. LP-CLIP [3] trains a linear layer using pseudo-labels produced by CLIP, enhancing robustness through knowledge distillation on the training set. In contrast, we train the student network on synthetic data via meta-learning. We explore applying LP-CLIP’s mixture of data augmentations and consistency loss to our student model distillation process. As shown in the table below, consistency loss achieves comparable performance but slightly lags in generalization across different datasets.
> |    |Caltech-101|ImageNet1|ImageNet2|ImageNet3|ImageNet4|Average|
> |:--:|:--:|:--:|:--:|:--:|:--:|:--:|
> | Ours | 61.33 | 62.46 | 65.02 | 65.60 | 62.52 |63.39|
> | LP-CLIP | 61.40 | 60.50 | 62.34 | 64.68 | 63.94 | 62.57 |
> ___
> [4] Waffling Around for Performance: Visual Classification with Random Words and Broad Concepts. ICCV 2023.
> [5] Denoising Diffusion Implicit Models. ICLR 2021.
> [6] Pseudo Numerical Methods for Diffusion Models on Manifolds. ICLR 2022.
> [7] On First-Order Meta-Learning Algorithms. ArXiv 2018.

---

> ### Comment · Reviewer_Radj · 2024-11-27
>
> Thanks to the authors for rebuttal for addressing my concerns. The explanations are satisfactory. I would like to increase my score to 8.

---

### Official Review · Reviewer_WBiv · 2024-11-04

**Soundness:** 3
**Presentation:** 3
**Contribution:** 3
**Rating:** 6
**Confidence:** 4

**Summary:**

This paper delves into Data-Free Knowledge Distillation for CLIP so as to distill a compact student model with customized zero-shot or few-shot image classification capacity. Specifically, the proposed framework is composed of surrogate dataset generation and knowledge distillation. For the former component, this paper uses model inversion and style dictionary diversification based on the framework of VQGAN-CLIP. For the latter component, this paper designs a meta method for knowledge distillation. Experiments validate the effectiveness of the proposed framework.

**Strengths:**

-	This paper is well written.
-	This paper is well motivated to study DFKD for vision-language foundation models.
-	Experiments show the effectiveness of the proposed framework.

**Weaknesses:**

- My concern mainly lies in technique novelty. In Fig.1, the proposed framework is composed of dataset inversion process and knowledge distillation process. However, in dataset inversion process, the proposed method is mainly similar to [1] and [2], especially [2], which is also a related work to study DFKD in CLIP. In knowledge distillation, the proposed method is mainly similar to [3], which uses a MAML-like meta learning to enhance cross-domain generalization capacity.

[1] VQGAN-CLIP: Open domain image generation and editing with natural language guidance. ECCV 2022.

[2] Distilling vision-language foundation models: A data-free approach via prompt diversification. ACMMM 2023.

[3] Learning to Generalize: Meta-Learning for Domain Generalization. AAAI 2018.

**Questions:**

My concern mainly lies in technique novelty. Can you summarize your contribution again based on my concern?

---

> ### Author Response · Authors · 2024-11-19
> **Author Rebuttal (1/1)**
>
> > Q: My concern mainly lies in technique novelty. Can you summarize your contribution again based on my concern?
>
> A: Our work addresses real-world customization scenarios where users upload required texts or example images to meet practical tasks. We focus on improving the realism and consistency of synthetic images, expanding distributions based on few-shot images, and leveraging synthetic data properties to enhance the generalization of knowledge distillation. Below is a summary of our contributions based on your concern:
>
> For dataset inversion：
> * Synthetic images from [1] often lean toward artistic styles due to CLIP's training on internet data (see Figure 4). To address this, we redesign the style dictionary to better suit real-world applications. Experiments using the style dictionary from [2] reveal a saturation trend, where irrelevant styles introduce negative bias. In comparison, our strategy balances realism while maintaining diversity through optimized style prompts.
>   | |Caltech-101|
>   |:--:|:--:|
>   |16 styles|57.99|
>   |50 styles|60.78|
>   |86 styles|60.49|
>   |Our 16 styles|61.33|
> * We propose class consistency maintaining to **prevent overly stylized deviations**, which introduces and generates additional semantic information. The classification head, constructed with class semantics, acts as an anchor to regularize diversified data within CLIP’s embedding space. This method yields an average improvement of 0.6%.
> * We innovatively propose image-based customization for inversion, constructing prototypes from example images to **reduce intra-class variance**. Instead of solely relying on few example images for knowledge distillation, we expand the distribution by leveraging the teacher model’s knowledge. This approach effectively reduces generalization error and enhances performance.
>   ||Image-based|Text-based|Improvement|
>    |:---:|:---:|:---:|:---:|
>    |Caltech-101|84.78|61.33|23.45|
>    |ImageNet|74.54|65.15|9.39|
>    |Flower-102|74.72|18.07|56.65|
>
> Based on your suggestion, we revised the manuscript structure to better situate the contributions. *Preliminary* is now presented as Section 3. The first part rethinks BN-based model inversion, while the second part introduces the VQGAN-CLIP inversion paradigm. We also moved style dictionary diversification details into this section for better clarity.
>
> For knowledge distillation:
>
> Synthetic data inherently challenges generalization due to covariate shifts. To address this, we employ meta knowledge distillation on synthetic data with diverse styles. Our contribution lies in complementarily leveraging meta-learning and style diversification for data-free knowledge distillation. Meta-learning implicitly optimizes gradient inner product [4], a widely recognized technique for enhancing generalization [3,5,6]. By integrating this into our framework, we improve the effectiveness of knowledge distillation.
> ___
> [4] On First-Order Meta-Learning Algorithms. ArXiv 2018.
> [5] Gradient Matching for Domain Generalization. ICLR 2022.
> [6] Implicit Gradient Alignment in Distributed and Federated Learning. AAAI 2022.

---

> > ### Comment · Reviewer_WBiv · 2024-11-28
> >
> > After reviewing the paper again, I found this paper only cites the previous work [2] in a few inconspicuous places without discussing the comparison with [2], even not appearing in the section of related works. For example, in the section of related works, this paper claims that this paper focus on VLMs compared with previous DFKD works. But [2] is the first work to study DFKD in VLMs. Considering the rebuttal addresses my concerns to some extents, I can only increase the original score to 6.
> >
> > [2] Distilling vision-language foundation models: A data-free approach via prompt diversification. ACMMM 2023.

---

> ### Author Response · Authors · 2024-11-28
>
> Apologies for missing the PDF upload deadline by just a few minutes—we had already finalized the revised manuscript, but unfortunately, it was slightly late.
>
> In the DFKD paragraph of the related work section, we primarily discussed BN-based DFKD methods such as DeepInv, CMI, and Fast to introduce the subsequent Rethinking Model Inversion. Meanwhile, [2] and image-text matching were discussed in the introduction and preliminary sections.
>
> Addressing your concern, we have revised the DFKD paragraph in the related work section as follows:
>
> *... They typically assume a classification model with a classifier that outputs logits. Target classes are represented as labels defined in the classifier's label space.* (Original content introducing DeepInv, CMI, etc.) *In contrast, [2] proposed leveraging image-text matching for DFKD in VLMs, introducing three prompt diversification methods to extract out-of-distribution capacity from VLMs. In comparison, our target classes can be represented not only as texts but also as example images. Additionally, we propose consistency strategies to prevent noise semantics introduced by styles and enhance realism.*
>
> This revised manuscript will be uploaded in the final version. Thank you again for your suggestions.

---

### Official Review · Reviewer_HWxX · 2024-11-10

**Soundness:** 4
**Presentation:** 3
**Contribution:** 4
**Rating:** 10
**Confidence:** 3

**Summary:**

The paper shows that the existing works that use DFKD methods using CLIP do not perform well. This is blamed on their use of BatchNorm that biases towards faces. The paper introduces an alternate technique for performing DFKD well using CLIP - a text-image matching technique.

**Strengths:**

This paper presents a novel approach to open-vocabulary customization of Vision-Language Models (VLMs) like CLIP. The authors identify the limitations of existing Data-Free Knowledge Distillation (DFKD) methods and propose a novel solution to address these limitations.

The paper is well-written and easy to follow. The authors provide a clear motivation for their work and a concise overview of their proposed technique. The introduction effectively sets the stage for the paper, with a clear articulation of the research gap and the proposed solution.

The authors provide a comprehensive analysis of related work, demonstrating the novelty of their approach. The experimental findings are compelling, particularly the observation that CLIP's BN layers tend to favor faces, highlighting their unsuitability for DFKD.

The proposed framework's ability to handle both text-based and image-based customization enhances its applicability and significance. The use of instance-level contrastive loss for increased diversity is well-justified, both in practice and through theoretical analysis (Theorem 4.1).

The experimental setup and training details are described thoroughly, which is commendable. The choice of the ImageNet dataset is appropriate, given its scale and diversity. The result analysis is comprehensive and insightful, with the authors exploring various aspects of their approach, including the unique "warm-up" strategy.

**Weaknesses:**

1. While Figure 1 provides a good overview of the framework, consider replacing the "frozen" and "not frozen" symbols with more intuitive icons, such as a lock and an unlocked lock. Additionally, ensure the frozen symbol is clearly visible in the blue boxes, perhaps by changing its color.
2. Tables 2, 4, and 5 don’t have any units for the numbers or any text mentioning the metric used for those results. Please consider adding metrics and units.

**Questions:**

1. Why did you use VQGAN? Will the generated data have enough diversity? What other architectures did you consider?

---

> ### Author Response · Authors · 2024-11-19
> **Author Rebuttal (1/1)**
>
> > Q1: While Figure 1 provides a good overview of the framework, consider replacing the "frozen" and "not frozen" symbols with more intuitive icons, such as a lock and an unlocked lock.
>
> A1: Thank you for your feedback. We have updated the symbols in Figure 1, replacing the "frozen" and "not frozen" symbols with more intuitive icons to enhance visual clarity.
> ___
> > Q2: Tables 2, 4, and 5 don’t have any units for the numbers or any text mentioning the metric used for those results. Please consider adding metrics and units.
>
> A2: The numbers in Tables 2, 4, and 5 represent the classification accuracy (in %) of the student model. Thanks for your suggestions. We have clarified the metric in the experimental setup and added the units to the tables for consistency and clarity.
> ___
> > Q3: Why did you use VQGAN? Will the generated data have enough diversity? What other architectures did you consider?
>
> A3: Directly optimizing in the image $\hat{\boldsymbol{x}}$ pixel space is possible, but images typically reside in high-dimensional spaces (e.g., 224 $\times$ 224 $\times$ 3), making the optimization problem challenging and computationally expensive. To achieve more efficient parameterization, we adopt the pre-trained VQGAN decoder and perform optimization in the latent variable space $\boldsymbol{z}$, which is low-dimensional. Compared with the pixel updating strategy that updates different pixels independently, the generator can provide stronger regularization on pixels since they are produced from shared weights.
>
> We conduct the following experiments: directly inversing images without VQGAN, using VQGAN pre-trained on ImageNet (the VQGAN used in our paper), using VQGAN pre-trained on OpenImages. We also experimented with training VQGAN from scratch during the inversion process but encountered severe mode collapse.
> | |ImageNet1|ImageNet2|
> |--:|:-:|:-:|
> |Without VQGAN|34.24|33.60|
> |ImageNet VQGAN|62.46|65.02|
> |OpenImages VQGAN|59.36|63.68|
>
> As shown, using ImageNet VQGAN performs better on ImageNet than OpenImages VQGAN. The w/o VQGAN involves optimizing pixels directly for 400 iterations. While optimizing for thousands of iterations (as done in DeepInv) could improve results, this highlights the efficiency advantage of VQGAN.
>
> We also considered using diffusion models as generators. Unlike VQGAN, which maps $\boldsymbol{z}$ to the image $\hat{\boldsymbol{x}}$ in a single forward pass, diffusion models require a multi-step denoising process to generate images from noisy inputs or latent variables. Even advanced samplers like DDIM [1] or PNDM [2] typically require more than 10 steps. Performing model inversion involves optimizing input latent variables, which need to be **differentiable**. This means the full computation graph must be retained. However, most existing diffusion libraries (e.g., Hugging Face's `Diffusers`) use `no_grad()` during generation. To address this, we implemented a differentiable version of DDIM for experimentation, but found that even with just 5 steps, the computation graph exceeded 24GB of GPU memory. In summary, we choose VQGAN for its efficiency and effectiveness.
> ___
> [1] Denoising Diffusion Implicit Models. ICLR 2021.
> [2] Pseudo Numerical Methods for Diffusion Models on Manifolds. ICLR 2022.

---

> > ### Comment · Reviewer_HWxX · 2024-11-26
> >
> > Thank you for addressing the queries. The responses are satisfactory.

---

### Official Review · Reviewer_fKMo · 2024-11-10

**Soundness:** 3
**Presentation:** 2
**Contribution:** 3
**Rating:** 8
**Confidence:** 3

**Summary:**

The paper presents a novel approach for open-vocabulary customization in vision-language models like CLIP, utilizing Data-Free Knowledge Distillation. The authors address limitations of existing DFKD methods, which depend heavily on BatchNorm layers incompatible with CLIP. Their method incorporates image-text matching to invert a surrogate dataset, enabling text- and image-based customization. Key innovations include style dictionary diversification, class consistency maintaining, and meta knowledge distillation to enhance the generalizability of a student model.

**Strengths:**

The paper provides a meaningful contribution to open-vocabulary customization for VLMs, especially under data-free constraints. It addresses practical issues in adapting CLIP without original data, proposing a unique approach to handle limitations posed by BatchNorm layers. Techniques like style dictionary diversification and meta knowledge distillation are well-conceived, though the performance improvements are modest. While the theoretical analysis is detailed, the practical gains might benefit from further validation. Overall, the paper offers useful insights but may require more refinement and broader evaluation to strengthen its impact.

**Weaknesses:**

The paper's writing could be improved for clarity, as the relevance of BatchNorm (BN) statistics to the later-introduced contrastive learning method is somewhat confusing. The presentation would benefit from clearer contextualization and integration with recent advancements in VLM customization to help situate the contributions more effectively. While the proposed techniques are valuable, additional clarity around specific limitations—such as the potential for style dictionary diversification to introduce noise—could strengthen the paper. Additionally, the reliance on the CLIP model may limit generalizability across other VLM architectures. Expanding future work to include broader applications of the method across diverse vision-language architectures would help validate its adaptability.

**Questions:**

1. Could the authors elaborate on potential methods to mitigate noise introduced by style dictionary diversification, especially in fine-grained tasks?
2. Are there specific aspects of CLIP’s architecture that are essential to this approach, or could it be adapted to other VLM architectures?
3. In Figure 6, the style differences are not very apparent—could the authors clarify how style diversification manifests visually?

---

> ### Author Response · Authors · 2024-11-19
> **Author Rebuttal (1/2)**
>
> > Q1: The relevance of BatchNorm (BN) statistics to the later-introduced contrastive learning. The presentation would benefit from clearer contextualization and integration with recent advancements in VLM customization.
>
> A1: Previous studies rely on BN statistics but fail when applied to VLMs. In response, our *Preliminary* introduces an alternative inversion way: **image-text matching**, which does not require BN. Additionally, Appendix C explains why image-text matching outperforms BN-based methods. BN-based CMI [1] uses contrastive learning to ensure that images are distinguishable from previously synthesized images (line 125). In contrast, we use contrastive learning to enhance the diversity of text prompts, thereby increasing the diversity of synthesized images.
>
> Inspired by your suggestions, we have revised the structure to better situate our contributions. The *Preliminary* section is now established as Section 3. The first part discusses our rethinking of existing BN-based model inversion, while the second part introduces the image-text matching advancements upon which our work is based. Following this contextualization in the *Preliminary* section, we transition into the Methodology section.
> ___
> > Q2: Are there specific aspects of CLIP’s architecture that are essential to this approach, or could it be adapted to other VLM architectures?
>
> A2: Our method is applicable to VLMs with a vision encoder and text encoder structure. We also conducted inversion experiments on PyramidCLIP (which outperforms CLIP by 10%) and found that the issue of unusable BN layers is widespread; please refer to Appendix B for details. Based on your suggestion, we have also included the performance of our method on BLIP [2] and EVA [3] in the revised manuscript.
> |    |Caltech-101|ImageNet1|ImageNet2|ImageNet3|Average|
> |:--:|:--:|:--:|:--:|:--:|:--:|
> | CLIP | 61.33 | 62.46 | 65.02 | 65.60 | 63.60 |
> | BLIP | 59.68 | 50.88 | 57.62 | 57.36 | 56.39 |
> | EVA | 67.62 | 65.24 | 66.96 | 66.42 | 66.56 |
>
> As shown in the table, EVA achieves the best performance, followed by CLIP and BLIP. The pre-trained weights we use for BLIP are the official `blip-itm-base-coco`. We observe that BLIP’s image-text matching capability is weaker than CLIP's. This is because BLIP's text encoder is image-grounded and trained on a binary classification task conditioned on images. Additionally, BLIP’s pre-training dataset contains 14M images, whereas CLIP is pre-trained on 400M text-image pairs. EVA achieves the highest zero-shot top-1 accuracy on ImageNet-1K, and demonstrates strong image-text matching capability. These results indicate that our method is applicable to other VLM architectures and that its performance improves with the capability of the underlying VLM.
> ___
> >Q3: While the proposed techniques are valuable, additional clarity around specific limitations—such as the potential for style dictionary diversification to introduce noise—could strengthen the paper.
>
> A3: The potential for style diversification to introduce noise is clarified in both the methodology and experiments. To mitigate the potential noise introduced by style diversification, we propose class consistency maintaining (lines 270-283), which promotes consistency between synthetic images and their corresponding class text. It serves as an anchor to regularize the class semantics of the diversified data within CLIP’s embedding space. Table 1 demonstrates that the combination of style dictionary diversification and class consistency maintaining achieves the highest performance improvement. Furthermore, Table 4 provides an ablation study on consistency and diversity, illustrating the trade-off between the two.
> ___
> [1] Contrastive Model Inversion for Data-Free Knowledge Distillation. IJCAI 2021.
> [2] BLIP: Bootstrapping Language-Image Pre-training for Unified Vision-Language Understanding and Generation. ICML 2022.
> [3] EVA: Exploring the Limits of Masked Visual Representation Learning at Scale. CVPR 2023.

---

> ### Author Response · Authors · 2024-11-19
> **Author Rebuttal (2/2)**
>
> > Q4: In Figure 6, the style differences are not very apparent—could the authors clarify how style diversification manifests visually?
>
> A4: Style diversification is achieved by encouraging diverse prompts while retaining the core class semantics. These prompts lead to subtle but meaningful variations across images within the same class. For example, in the "Water Lily" and "Airplane" categories, differences in color tones and lighting effects can be observed. For "Pyramid," variations appear in surface textures, while in the "Valley" category, the viewing angles differ.
>
> The style dictionary we devise aims to closely emulate real-world scenes. The selected words do not introduce additional semantics. We also experimented with a style dictionary containing styles sourced from the Internet and varied the number of styles.
>  | |Caltech-101|
>   |:--:|:--:|
>   |16 styles|57.99|
>   |50 styles|60.78|
>   |86 styles|60.49|
>   |Our 16 styles|61.33|
>
> Generally, increasing the number of styles enhances diversity. However, as shown in the table, we observe a saturation trend where irrelevant styles may introduce negative bias. This occurs because some styles inadvertently add class-irrelevant semantics, such as unrelated elements or complex backgrounds. To address this, we retain only words that describe photorealism and propose class consistency maintaining to prevent overly stylized deviation.

---

> ### Author Response · Authors · 2024-12-02
> **Gentle Reminder**
>
> Dear Reviewer fKMo,
>
> Thank you for your thoughtful feedback on our paper! We have done our best to address your concerns and questions. We would greatly appreciate it if you could let us know whether our response has addressed your concerns.
>
> We look forward to hearing from you.

---

> > ### Comment · Reviewer_fKMo · 2024-12-03
> >
> > Thanks authors for addressing my questions. I have raised my score to 8.

---

### Author Response · Authors · 2024-11-25
**Gentle Reminder**

We greatly appreciate all the reviewers' efforts in evaluating our work. We are encouraged by the positive comments regarding our clear motivation (`fKMo`, `HWxX`, `WBiv`, `Radj`), theoretical analysis (`fKMo`, `HWxX`), and extensive experiments (`HWxX`, `WBiv`, `Radj`). We benefit from the reviewers' constructive suggestions and have revised our manuscript accordingly. Additional analyses and experiments are included in Appendix, with revisions highlighted in blue.

We hope our responses address your concerns appropriately and welcome further discussions. Thank you again for your time and helpful comments. Have a good day!

---

### Meta-Review · Area_Chair_rx7Z · 2024-12-17

**Metareview:**

This paper introduces using image-text matching for Data-Free Knowledge Distillation (DFKD) for the CLIP model, which involves creating a surrogate dataset for distillation. This dataset is created to maintain diversity and class consistency. Comprehensive experiments show the effectiveness of this approach compared to others.

Weaknesses such as writing clarity, paper notations, and the use of VQGAN instead of diffusion, were already addressed by the authors during the rebuttal. Based on the strong results and the novelty of the approach, I would therefore recommend acceptance of the work.

**Additional Comments On Reviewer Discussion:**

The motivation of the paper is well-received by all the reviewers.

Reviewers fKMo and HWxX mentioned that writing and notations can be improved, which are addressed by the authors.

Reviewers fKMo and Radj mentioned the application of the approach to other CLIP variants. Reviewers HWxX and Radj question the use of VQGAN and the possibility of adopting diffusion models. Reviewer fKMo asked about the details of style diversification. All these questions are well answered by the authors by additional experiments and justifications.

---

### Decision · Program_Chairs · 2025-01-22

Accept (Oral)